# The contribution of Saharan dust to the ice nucleating particle concentrations at the High Altitude Station Jungfraujoch (3580 m a.s.l.), Switzerland

Cyril Brunner[1], Benjamin T. Brem[2], Martine Collaud Coen[3], Franz Conen[4], Maxime Hervo[3], Stephan Henne[5], Martin Steinbacher[5], Martin Gysel-Beer[2], and Zamin A. Kanji[1]

[1]Institute for Atmospheric and Climate Science, ETH, Zurich, CH-8092, Switzerland
[2]Laboratory of Atmospheric Chemistry, Paul Scherrer Institut, CH-5232 Villigen PSI, Switzerland
[3]Federal Office of Meteorology and Climatology, MeteoSwiss, CH-1530 Payerne, Switzerland
[4]Department of Environmental Sciences, University of Basel, CH-4056 Basel, Switzerland
[5]Laboratory for Air Pollution / Environmental Technology, Empa, Überlandstrasse 129, CH-8600 Dübendorf, Switzerland

**Correspondence:** C. Brunner (cyril.brunner@env.ethz.ch) and Z.A. Kanji (zamin.kanji@env.ethz.ch)

**Abstract.** The ice phase in mixed-phase clouds has a pivotal role in global precipitation formation as well as for Earth's radiative budget. Above 235 K, sparse particles with the special ability to initiate ice formation, ice nucleating particles (INPs), are responsible for primary ice formation within these clouds. Mineral dust has been found to be one of the most abundant INP in the atmosphere at temperatures colder than 258 K. However, the extent of the abundance and distribution of INPs remains largely unknown. To better constrain and quantify the impact of mineral dust on ice nucleation, we investigate the frequency of Saharan dust events (SDEs) and their contribution to the INP number concentration at 243 K and at a saturation ratio with respect to liquid water ($S_w$) of 1.04 at the High Altitude Research Station Jungfraujoch (JFJ; 3580 m a.s.l.) from February to December 2020. Using the single scattering albedo Ångström exponent retrieved from a nephelometer and an aethalometer, satellite retrieved dust mass concentrations, simulated tropospheric residence times, and the attenuated backscatter signal from a ceilometer as proxies, we detected 26 SDEs, which in total contributed to 17 % of the time span analyzed. We found every SDE to show an increase in median INP concentrations compared to that of all non-SDE periods, however, not always statistically significant. Median INP concentrations of individual SDEs spread between 1.7 and 161 INP std L$^{-1}$, thus, two orders of magnitude. In the entire period analyzed, 74.7 $\pm$ 0.2 % of all INPs were measured during SDEs. Based on satellite retrieved dust mass concentrations, we argue that mineral dust is also present at the JFJ outside of SDEs, but at much lower concentrations, thus still contributing to the INP population. We estimate 97 % of all INPs active in the immersion mode at 243 K $S_w$ = 1.04 at the JFJ to be dust particles. Overall, we found INP number concentrations to follow a leptokurtic log-normal frequency distribution. We found the INP number concentrations during SDEs to correlate with the ceilometer backscatter signals from a ceilometer located 4.5 km north of the JFJ and 1510 m lower in altitude, thus scanning the air masses at the same altitude as the JFJ. Using the European ceilometer network allows to study the atmospheric pathway of mineral dust plumes over a large domain, which we demonstrate in two case studies. These studies showed that mineral dust plumes form ice crystals at cirrus altitudes, which then sediment to lower altitudes. Upon sublimation in dryer air layers, the

residual particles are left potentially pre-activated. Future improvements to the sampling lines of INP counters are required to study if these particles are indeed pre-activated, leading to larger INP number concentrations than reported here.

## 1 Introduction

Sixty-three $\pm$ 7 % of global precipitation is initiated via the ice phase (Heymsfield et al., 2020), predominately over land and in the midlatitudes (Mülmenstädt et al., 2015). The co-existence of ice and metastable supercooled liquid water in a cloud is important for precipitation formation, as ice crystals grow efficiently at the expense of evaporating cloud droplets due to the lower equilibrium saturation vapor pressure over ice than over liquid water below 273 K (Wegener, 1911; Bergeron, 1935; Findeisen, 1938; Koop and Mahowald, 2013). Mixed-phase clouds, containing both the liquid and the ice phase, are topics of ongoing research to better constrain precipitation formation in climate and weather models. Clouds also have a special relevance to Earth's climate. Not only do clouds cover 68 % of Earth's surface (Stubenrauch et al., 2013), the phase of a cloud also strongly influences its radiative properties (e.g., Sun and Shine, 1994; Lohmann and Feichter, 2005), emphasising the need to adequately simulate cloud glaciation in climate models. Mixed-phase clouds theoretically can exist between 273 K and $\sim$ 235 K. Depending on the measurement location, in-situ measurements revealed that only approximately half of the clouds contain the liquid phase when at 253 to 258 K, while the warmer clouds are mostly ice free (e.g., Korolev et al., 2003; Verheggen et al., 2007; Kanitz et al., 2011). Below 235 K, cloud droplets freeze homogeneously. The stochastic process of solid cluster formation within the suspended liquid at atmospheric relevant sizes and timescales statistically favors the formation of stable ice clusters at these low temperatures. Between 273 K and $\sim$ 235 K, heterogeneous nucleation on ice nucleating particles (INP) is responsible for the primary ice formation within clouds (Vali, 1985; Vali et al., 2015). However, the abundance, sources and nature of INPs remains poorly understood (Murray et al., 2021).

Different atmospheric particles can act as INPs, depending on the ambient temperature and water vapor saturation. Mineral dust particles are a major constituent of atmospheric aerosols and dust plumes are transported over long distances between continents (Prospero, 1999). Transport of Saharan dust to Europe and the alpine region is a frequent phenomenon driven by the latitudinal movement of the prevailing large-scale circulation, including the movement of the Intertropical Convergence Zone. Furthermore, the North Atlantic Oscillation was found to modulate the mean occurrence of these weather patterns and, hence, Saharan dust transport to Europe (Moulin et al., 1997). Height-resolved global distribution lidar measurements showed that spring is the season with most dust plumes in the northern midlatitudes, with highest dust mass found in summer between 2000 and 3000 m a.sl. and in winter between 1000 and 2000 m a.s.l. (Liu et al., 2008). Wegener (1911) had the hypothesis of minerals being an important INP species, which, subsequently, were studied in more detail in multiple field (e.g., Sassen et al., 2003; Schrod et al., 2017, and references therein) and laboratory studies (e.g., Mason and Maybank, 1958; Field et al., 2006; Welti et al., 2009; Boose et al., 2016c). In these studies, mineral dust showed the ability to be ice-active in the deposition or immersion mode with concentrations exceeding 500 to 1000 INP std L$^{-1}$ at 236.6 to 248 K (DeMott et al., 2003, 2009; Bi et al., 2019). Zhao et al. (2021) studied the global contributions to INP concentrations at 248 K using the Community Earth System Model version 2 (CESM2) and found dust over the terrestrial midlatitudes to be the dominating INP species by one to two orders

of magnitude higher INP number concentrations at 248 K compared to marine organic aerosols. In general, the contribution of different INP species is expected to vary depending on the implementation of the INP parameterization in the model and aerosol representations in the model. Mineral dust has been found to be the most abundant INP species in the atmosphere at temperatures colder than 258 K (Hoose and Möhler, 2012; Murray et al., 2012), however, the contribution of mineral dust to the total INP population remains unknown. Studies were conducted at the High Altitude Research Station Jungfraujoch (JFJ), a mountain top station often located in the lower free troposphere, to assess, amongst others, the INP concentrations during Saharan Dust events (SDEs). Chou et al. (2011) showed that the INP number concentration can increase by one to two orders of magnitude during a SDE, indicating that SDEs have different intensities. Conen et al. (2015) studied the INP concentrations at the JFJ over one year with an offline technique and found a weak influence of Saharan dust events on the INP concentration at 265 K. Lacher et al. (2018a) analysed data from 9 individual field campaigns from winter, spring, and summer in the years 2014 - 2017 and quantified the INP concentrations at 241 - 242 K and at a saturation ratio with respect to liquid water of $S_w$ = 1.04. Yet, these field campaigns with a duration of up to 6 weeks were targeted specifically for periods with SDEs, therefore potentially introducing a bias to the overall quantification. In addition, satellite retrieved dust measurements demonstrate that the presence of atmospheric dust is not a binary phenomena, as the term SDE would imply, but in fact the dust concentrations show a strong temporal and spatial variation (e.g., Voss and Evan, 2020).

In this work, we investigate and quantify the INP concentrations at 243 K and $S_w$ = 1.04 (immersion freezing). These conditions were chosen to align with previous INP measurements at the JFJ between 2014 and 2017 (Boose et al., 2016a; Lacher et al., 2018a). Ice formation in stratiform mixed-phase clouds is frequently observed close to the cloud top where temperatures of 243 K are a common lower bound cloud top temperature of mixed-phase clouds in central Europe (e.g., Bühl et al., 2016). In addition, 243 K is the warmest temperature where the instrument's signal-to-noise ratio allows for statistically acceptable data analysis when the sampling site is located in a remote region such as mountain top stations or the Arctic without using an aerosol concentrator. There is an uncertainty of relative humidity and variation in the vertical position of the particles within the aerosol layer in the chamber (DeMott et al., 2015; Brunner and Kanji, 2021) amounting to $S_w$ + 0.007 and - 0.009 and ± 1.11 K at 243 K and set $S_w$ of 1.04. To ensure that the entire sample layer experiences $S_w$ > 1.0, a nominal $S_w$ = 1.04 was chosen. All INP concentrations were measured at the JFJ during all SDEs between February 7 and December 31, 2020. During this time, continuous high-resolution (20 minutes) online INP measurements were performed for the first time at the JFJ. Because the data are not tied to single field campaigns in active SDE seasons, it also includes measurements in seasons where SDEs are infrequent. This allows to analyze whether all SDEs show an increased INP number concentration, as previous studies imply (Chou et al., 2011; Boose et al., 2016b; Lacher et al., 2018a). Furthermore, the classification of SDEs is based on four distinct tracers (see Section 2.2) and analyzed with regard to the type of air mass present at the site, i.e., free tropospheric air or boundary layer intrusions (see Section 2.4). Our data indicate that signals from Light Detection and Ranging (LIDAR) ceilometers can be used to infer INP concentrations, as reported in other studies using depolarization channel LIDARs (Mamouri and Ansmann, 2015; Ansmann et al., 2019). In contrast to Mamouri and Ansmann (2015) and Ansmann et al. (2019), the topographic setup of the present study allowed for the ceilometer to scan the same altitude that the INP concentrations were measured at. Estimating the INP concentrations from the ceilometer backscatter signals from all

90 ceilometer stations across Europe allows us to (back-)track the aerosol masses with enhanced INP concentrations and look into their atmospheric pathway, which we demonstrate in a case study. Finally, the contribution of (Saharan) dust to the INP concentration is estimated.

## 2 Materials and methods

### 2.1 Site description

The Sphinx observatory at the JFJ is located on a saddle between Mt. Mönch and Mt. Jungfrau in the Swiss alps (46.330° N, 7.590° E) at an altitude of 3580 m a.s.l. (see Figure 1). Due to its altitude, the site experiences free tropospheric (FT) air masses, mainly in winter and during night time, as well as boundary layer intrusions (BLI), predominantly in summer and during day time (Collaud Coen et al., 2011; Herrmann et al., 2015). The local wind directions measured at the Jungfraujoch are strongly driven by the topography around the Jungfraujoch and are not representative for the larger scale wind direction at
this altitude level (see Figure A1 in the Appendix, and e.g., Ketterer et al. (2014)). The principal local wind directions between February and December 2020 were 320° (NW) for 62 %, and 150° (SE) for 27 % of the time, while calm wind situations below 1 m s$^{-1}$ had a frequency of 11 %. To the northwest there is a steep drop of more than 1500 m, whereas to the southeast the elevation decreases steadily over the Aletsch glacier. The JFJ has hosted long-term aerosol measurements for more than 30 years and served as a platform for many previous studies to investigate physical, chemical, and optical aerosol properties as
well as aerosol-cloud interactions and cloud characteristics (see e.g., Bukowiecki et al., 2016). There is no appreciable natural source of mineral dust in proximity of the site. Potential local sources of arable dust are isolated agricultural fields 15 km north of the JFJ.

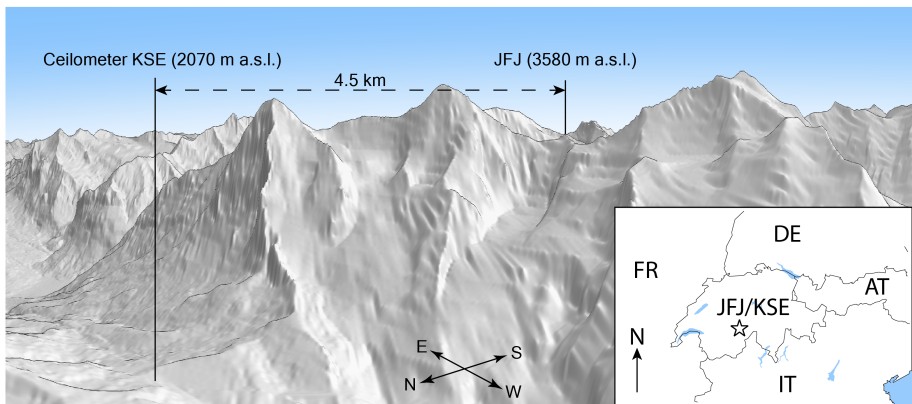

**Figure 1.** Illustration of the location of the INP sampling location at the JFJ and the ceilometer at the Kleine Scheidegg (KSE) embedded within the topography and in Switzerland. The topography was extracted from the digital height model DHM25 from the Federal Office of Topography swisstopo.

## 2.2 Classification of SDEs

In a first step, SDEs were identified using four distinct tracers, the ceilometers in close proximity of the JFJ (at the Kleine Scheidegg, KSE), the single scattering albedo Ångström exponent ($\alpha_{SSA}$) measured at the JFJ, dust retrieved from satellite remote sensing generated using the Copernicus Atmosphere Monitoring Service ($dust_{CAMS}$), and FLEXPART-particle residence times over the Saharan region. The methodology of all tracers is introduced in the upcoming dedicated subsections. If all tracers show a signal, the event is classified as high confidence Saharan dust event (hcSDE). If at least one of the tracers shows a signal, but not all tracers agree, then the event is classified as low confidence Saharan dust event (lcSDE). The beginning and end of each SDE reflect the beginning and end of the onset of the earliest tracer or the decay of the latest tracer, respectively. If none of the tracers shows a signal, the event is classified as non-Saharan dust event (non-SDE). Only after the classification of SDEs the INP measurements at the JFJ are analyzed.

### 2.2.1 Ceilometer

The Lufft CHM15k-Nimbus ceilometer is operated by MeteoSwiss at the KSE (46.547° N, 7.985° E, 2070m a.s.l.), 4.5 km north of the JFJ and 1510 m lower in altitude (see Figure 1). With a 5° zenith angle, it also scans the air mass at the same altitude as the JFJ, retrieving information about the vertical and temporal evolution of aerosol plumes at these altitudes with a temporal resolution of 5 min. More detailed information about the ceilometer and the site can be found in Hervo et al. (2016). The ceilometer data are qualitatively analysed for aerosol plumes. If an aerosol plume was detected above the planetary boundary layer (PBL), the period for which the plume is observed between 3200 - 4000 m a.s.l. is marked as a SDE. In case the signal was strongly attenuated by clouds or precipitation such that a potential signal from an aerosol plume was masked, i.e., the attenuated backscatter was $> 10$ m$^{-1}$ sr$^{-1}$, the corresponding period is labeled as "cloud/precipitation" and no SDE signal was retrieved until the signal is restored. If an aerosol plume was visible before and after periods with clouds or precipitation with a total duration $< 12$ hours, the total period including the periods with clouds or precipitation was marked as a SDE. The data are accessed using the Centre for Environmental Data Analysis (CEDA, Met Office, 2021).

### 2.2.2 Single scattering albedo Ångström exponent

The single scattering albedo Ångström exponent ($\alpha_{SSA}$) is an indicator of aerosol optical properties, which change during the presence of SDEs. Collaud Coen et al. (2004) observed that the exponent of the single scattering albedo during SDEs decreases with wavelengths, which counteracts the usual increasing trend. This was attributed to the different chemical composition of the mineral aerosol particles and their larger size. $\alpha_{SSA}$ is retrieved from a nephelometer (Airphoton, IN101) and an aethalometer (MAGEE scientific, AE33) according to Collaud Coen et al. (2004). Both instruments are run at the Sphinx observatory on the JFJ. A SDE is detected if the $\alpha_{SSA}$ is negative for more than 6 consecutive hours. This is longer than the previously used 4 hours in Collaud Coen et al. (2004) in order to decrease the number of false or suspicious signals due to construction work at the JFJ (see below).

### 2.2.3 Dust from CAMS

The daily Copernicus Atmosphere Monitoring Service (CAMS) air quality forecast retrieves, amongst others, an hourly updated dust product with a horizontal resolution of 0.1° and covers the European domain (25.0° W to 45.0° E, 30.0° N to 72.0°N). It is based on an ensemble of 9 state-of-the-art numerical air quality models developed in Europe: CHIMERE, EMEP, EURAD-IM, LOTOS-EUROS, MATCH, MOCAGE, SILAM, DEHM, and GEM-AQ (Giusti, 2021). Gueymard and Yang (2020) performed a worldwide validation of the aerosol optical depth and Ånsgtröm exponent from CAMS and MERRA-2 with ground-based

AERONET stations over the period 2003–2017. They found the root-mean-square error to vary in the range 0.031–0.268 for the aerosol optical depth in CAMS and 0.382 for the Ångström exponent. O'Sullivan et al. (2020) compared CAMS to in-situ and remote sensing measurements and found dust aerosol optical depth predictions to be generally in good agreement, but observed a low bias. The vertical location was at a lower altitude in CAMS than in observations and CAMS underpredicted the coarse-mode dust while overpredicting fine-mode dust.

In this work the CAMS data are generated using Copernicus Atmosphere Monitoring Service information (2020). It is important to note that neither the European Commission nor ECMWF is responsible for any use that may be made of the Copernicus information or data it contains. For the classification of SDEs we used the dust reanalysis data from CAMS ($dust_{CAMS}$) at 1000 m above the surface in hourly resolution and units of µg m$^{-3}$, accessed via the Copernicus Atmosphere Data Store. 1000 m above surface was chosen to be closest to the real altitude of the JFJ and accounting for the smoothed

surface elevations in the model domain due to the coarse grid spacing. The data was extracted for the closest grid point (46.55° N, 7.95° E) to the coordinates of the JFJ.

The $dust_{CAMS}$ concentrations show dust to be always present at the JFJ, thus requiring a threshold above which $dust_{CAMS}$ shows a positive SDE signal. This threshold can be set arbitrarily, resulting in fewer or more SDEs. Conventionally, dust events at the JFJ were defined by the $\alpha_{SSA}$ signal (e.g., Collaud Coen et al., 2004; Lacher et al., 2018b). We choose the

following procedure to tune the threshold to best agree with the onset and decay of the $\alpha_{SSA}$ signal. According to Ott (1990), when pollutants are measured at a point away from their sources within the troposphere, the observed frequency distribution of a given pollutant concentration is log-normally distributed because of the successive random dilution by large and small scale dynamics. This also applies to the $dust_{CAMS}$ concentrations at the JFJ, as illustrated in Figure 2. A Gaussian fit to the logarithm of the $dust_{CAMS}$ concentrations allows to calculate standard thresholds used in statistics, like the third quartile

plus 1.5 times the inter quartile range (Q3 + 1.5 IQR), a metric regularly used for whiskers in box plots. We found that the mean plus 1.5 times standard deviation ($\mu + 1.5\,\sigma$) agrees best historic SDE signals in $\alpha_{SSA}$ and can be used as a general threshold for $dust_{CAMS}$ concentrations above which a SDE is indicated. In this study, the threshold for a SDE signal translates to $dust_{CAMS} \geq \mu + 1.5\,\sigma = 2.36\,µg\ m^{-3}$.

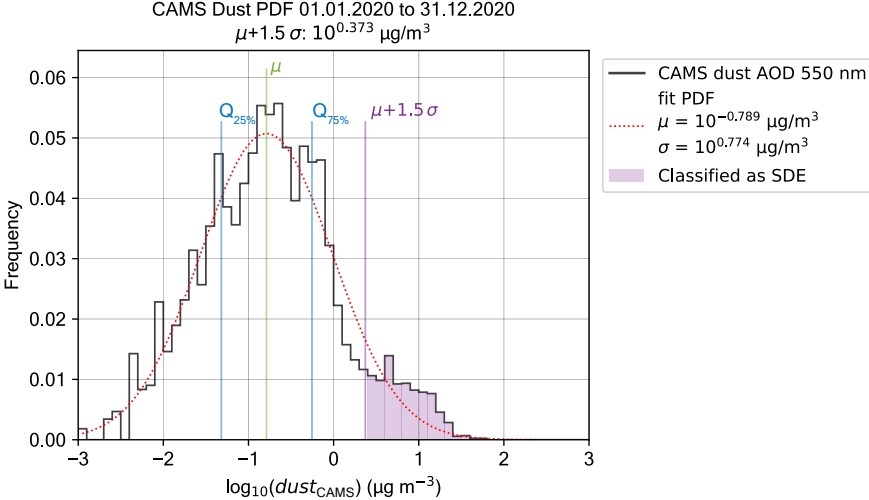

**Figure 2.** Frequency distribution of $dust_{CAMS}$ concentrations at the JFJ with fitted log-normal curve, the corresponding mean, $25^{th}$ and $75^{th}$ percentiles ($Q_{25\%}$ and $Q_{75\%}$) and the threshold for a SDE signal of $dust_{CAMS} \geq \mu + 1.5\,\sigma = 2.36$ µg m$^{-3}$, indicated in purple shading.

### 2.2.4 FLEXPART

The Lagrangian particle dispersion model (LPDM) FLEXPART (FLEXPART, Version 9.1_EMPA, Stohl et al., 2005) is used to estimate the surface residence times over the Saharan desert of air parcels arriving at the JFJ. The Saharan desert is here defined by composing the countries Algeria, Chad, Egypt, Libya, Mali, Mauritania, Morocco, Niger, Western Sahara, Sudan, and Tunisia. FLEXPART was driven by operational HRES analysis/forecast fields obtained from the ECMWF Integrated Forecasting System (IFS) with a 3-hourly temporal resolution and a spatial resolution of 0.2°×0.2° over the Alpine domain

and 1°×1° elsewhere. Simulations were done in receptor-oriented backward mode for an inert air tracer, releasing 50'000 particles every 3-hours at the location of Jungfraujoch (3100 m a.s.l. to account for smoothed topography) and tracing these back for 10 days in the atmosphere. Surface residence times (given in units time divided by local air density) were evaluated along the backward simulation and cumulated for the areas mentioned above. We note that due to the use of an inert tracer and the lack of a dust activation indicator in the potential Saharan source regions, the FLEXPART residence time indicator can only

give a qualitative indication of dust transport to JFJ. A signal for a SDE is present if the density weighted particle residence time exceeds background noise levels, here 100 s m$^3$ kg$^{-1}$.

### 2.3 INP measurements

The INP concentrations were measured with an autonomous continuous flow diffusion chamber, the automated Horizontal Ice Nucleation Chamber (HINC-Auto, Brunner and Kanji, 2021), at $T$ = 243.15 K and $S_w$ = 1.04 at the Sphinx observatory on

the JFJ. All INP concentrations are in units of INP std L$^{-1}$ (per standard liter, normalized to $T$ = 273.15 K and an atmospheric pressure of $p$ = 1013.25 hPa). Ambient air is sampled using a heated total aerosol inlet (293.15 K, Weingartner et al.,

1999), which also feeds other aerosol measurements (a condensation particle counter (CPC), a scanning mobility particle sizer (SMPS), an optical aerosol spectrometer (OAS, FIDAS 200, Palas GmbH, Germany), two nephelometers, two filter-based absorption photometers, and a particle mass monitor). Due to the low volume flow of HINC-Auto and the sampling line geometry, particle survival measurements using an optical particle counter (MetOne GT-526S) and an OAS showed limitations when sampling particles $d > 2.5$ µm (see Appendix A for detailed information), despite the fact that no impactor was used upstream of HINC-Auto. According to particle size distribution measurements from the OAS, particles with $d > 2.5$ µm contributed during the investigated SDE periods on average to 35 % to the overall particle surface area (see Figure A2 for an example of particle size distributions during a SDE). This is a noteworthy limitation, as studies have reported a majority of INPs at T > 248 K to be supermicron particles (e.g., Mason et al., 2016; Creamean et al., 2018; Gong et al., 2020), and for a constant ice active fraction INP concentrations scale with total particle surface area for a given temperature (e.g., Connolly et al., 2009; Niemand et al., 2012). However, instrument comparisons typically report substantially larger discrepancies between individual instruments than 35 % (e.g., see Hiranuma et al., 2015; DeMott et al., 2018).

INPs are detected if particles with an optical diameter of $\geq 4.0$ µm are counted at the chamber exit. Only particles with a diameter < 2.5 µm are sampled by the chamber, and the maximum expected size of a droplet that activated on a 2.5 µm particle is well below 4 µm. Therefore, the method is robust as only ice crystals grow past the set size threshold of 4 µm in the set conditions. The frequency of the INP measurements is every 20 min (15 min sampling plus 5 min background), corresponding to N = 19561 measurements between February 7 and December 31, 2020. See Brunner and Kanji (2021) for more information on the sampling and derivation of the INP concentrations. During the investigated time period, construction work was undergoing in the tunnels of the JFJ, as described in Bukowiecki et al. (2021). In case of local pollution, high frequency fluctuations in particle concentrations were observed (e.g., in the CPC measurements). To filter periods of local pollution, the following two methods were applied: Firstly, the smallest OPC size bin of HINC-Auto (0.3 µm) was analyzed with regard to fluctuations. A typical INP measurement includes 180 OPC sequences of 5 seconds each. For each 5 second sequence the cumulative number of particles with a optical size of $\geq 0.3$ µm is counted. If the count of a 5 second sequence is 30 % higher than the average count of all 180 sequences, the corresponding INP measurement is flagged. Secondly, fluctuations in the CPC measurements at the JFJ were analyzed. If the one-minute average concentration changed by more than 40 particles per cm$^3$ from the previous one-minute average, the current and the subsequent five minutes of measurements were flagged. These thresholds were chosen based on a qualitative assessment of the fluctuations in the OPC and CPC measurements. All INP measurements containing flagged CPC periods were flagged as well. Applying both methods is estimated to conservatively flag potentially polluted INP measurements and leads to N$_{unflagged}$ = 14216 (72 %) measurements unaffected by local pollution.

False-positive counts can arise in HINC-Auto. Frost grows on the chamber walls, breaks off and is detected at the outlet as ice. This happens irrespective of whether ambient or particle-free air is sampled with HINC-Auto. Thus, to correct the measured INP concentrations, the number of frost particles is measured separately and subtracted from the uncorrected INP measurements. This is done by sampling particle-free air for a period of 5 min before and after an ambient air measurement, particle-free air is sampled. During these periods, the number of false-positive frost particles are counted and subtracted time-proportional from the ambient air measurement in between. The recorded false-positive counts per unit time follow a Poisson

distribution. Therefore, in a fraction of cases more or fewer false-positive counts per unit time are recorded during the particle-free measurements than during the ambient air measurement. This results in fluctuations of the measured INP concentrations even if the true atmospheric INP number concentration was to remain constant. The standard deviation of the resulting prob-
ability density function corresponds to the stated counting uncertainty of $\pm 1\,\sigma$, which is provided with INP concentrations stated in the present work. This counting uncertainty is also considered to be the $1\,\sigma$ limit of detection ($1\sigma$-LOD) for a single data point. However, in the present work all background-corrected INP concentrations are retained, including positive values below the $1\sigma$-LOD and negative values. This approach ensures that the random noise in background-corrected INP values caused by subtracting the mean frost particle counts does not introduce a systematic bias in mean or median values, which
would occur if data below the $1\sigma$-LOD were discarded.

## 2.4   Classification of air mass

The decay of 222-Radon ($^{222}$Rn) is measured at the JFJ according to Griffiths et al. (2014). $^{222}$Rn is a tracer of BLI at high altitude sites, as there are no substantial sources of $^{222}$Rn in the FT. $^{222}$Rn is formed by the decay of naturally occurring radioisotopes in minerals. If the surrounding rock at a mountain site is covered by snow and ice, as is mostly the case at the
JFJ, the formed $^{222}$Rn is inhibited from mixing into the ambient air, leading to low $^{222}$Rn concentrations when the site is experiencing FT air masses. On the other hand, $^{222}$Rn is released relatively homogeneously in time and space from soils and gets well-mixed within the PBL, leading to higher $^{222}$Rn concentrations at the JFJ during periods of BLI from air with contact to bare soil. Further information can be found in Griffiths et al. (2014).

When looking at remote sensing data of the PBL over alpine terrain (e.g., Nyeki et al., 2000), the air masses present at the site
are not binary (either FT or BLI), but the PBL can gradually get mixed with FT air (Henne et al., 2004). Consequently, setting a binary threshold neither resembles an atmospheric process, nor represents the recorded data. Therefore, we use an approach where each $^{222}$Rn concentration corresponds to a probability of the sampled air to be of free tropospheric origin (as explained below). For this, the $^{222}$Rn concentrations at the JFJ were analysed between 01.01.2009 and 31.12.2020, as shown in Figure 3. Founded on the theory where the concentration of pollutants in the atmosphere follows a log-normal distribution (Ott, 1990),
two normal distributions were fitted so that the sum of them closely reproduces the observed bimodal distribution of the logarithm of the $^{222}$Rn concentrations. One distribution corresponds to $^{222}$Rn concentrations found in the FT (fit parameter: $\mu_{\text{FT}}$ = -0.139 Bq/std m$^3$, $\sigma_{\text{FT}}$ = 0.239 Bq/std m$^3$) and the other distribution to the concentrations measured during BLI (fit parameter: $\mu_{\text{BLI}}$ = 0.403 Bq/std m$^3$, $\sigma_{\text{BLI}}$ = 0.238 Bq/std m$^3$).

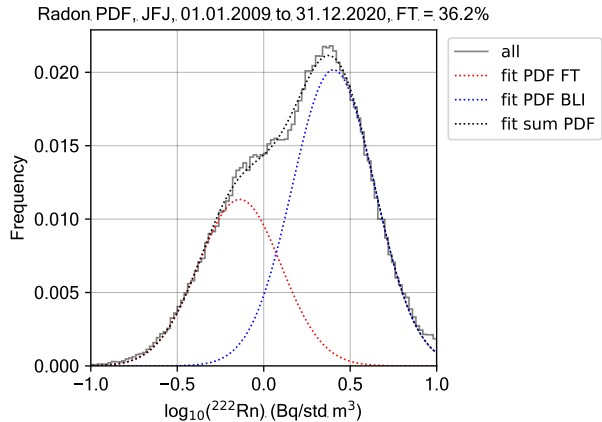

**Figure 3.** Frequency distribution of 222-Radon ($^{222}$Rn) concentrations at the JFJ between January 1, 2009 and December 31, 2020 in solid grey, with two fitted log-normal distributions, one for free tropospheric (FT, red dashed line) and the other one for boundary layer intruded (BLI, dashed blue) air mass, both scaled such that the sum (black dashed line) agrees best with the measured $^{222}$Rn frequency distribution. The fits allow to assess the probability of a given $^{222}$Rn concentration to be measured within FT air masses.

Besides $^{222}$Rn, the total number concentration of particles with diameters larger than $d \geq 90$ nm ($N_{90}$) is another tracer for BLI (Herrmann et al., 2015). The threshold above which the air mass is likely boundary layer intruded is $N_{90} \geq 100$ - 150 cm$^{-3}$, depending on the season (Herrmann et al., 2015). Ceilometer measurements can also be used to infer the top of the continuous aerosol layer, which indicates the top of the planetary boundary layer (Poltera et al., 2017), and subsequently, whether the JFJ is within the PBL. However, this method has not been used in the present study. A further tracer of BLI is the

ratio of total reactive nitrogen (NO$_y$) to carbon monoxide (CO) according to Zanis et al. (2007). Both NO$_y$ and CO are emitted from anthropogenic sources, however, NO$_y$ reacts and decays on the order of days, while CO can be considered inert within the same time period. Consequently, the ratio of NO$_y$ to CO decreases with increasing ageing time, leading to smaller ratios found in the FT compared to BLI (Zanis et al., 2007). CO is continuously measured with a Cavity Ringdown Spectrometer at the JFJ (Zellweger et al., 2019). NO$_y$ was measured until March 2020 with a chemiluminescence detector after conversion to

NO on a heated gold catalyst (573 K) in the presence of CO as a reducing agent (Pandey Deolal et al., 2012). An analysis of $^{222}$Rn, $N_{90}$ and NO$_y$/CO of data collected at the JFJ between 1. January 2017 and 31. December 2018 showed for $N_{90} \geq 100$ cm$^{-3}$ to be indicative of BLI, but concentrations of $N_{90} < 100$ cm$^{-3}$ to not always be indicative for FT air masses, as indicated by $^{222}$Rn and the NO$_y$/CO ratio (see Figure A3 in the Appendix). Therefore, in the present study a sigmoid function forces periods with concentrations of $N_{90}$ above a smooth threshold (to cover the seasonal variations of the threshold) to have a low

probability to be sampled within the FT, even if the $^{222}$Rn criterion indicates FT, but allows for periods with concentrations below the smooth threshold to be either FT or BLI depending on the $^{222}$Rn concentration. The threshold midpoint ($N_{90,\text{TH}}$) and smoothing, using the slope factor $k$ (here $N_{90,\text{TH}} = 120$ cm$^{-3}$ and $k = 0.1$) have been chosen to best agree with season-specific thresholds found in the literature (Herrmann et al., 2015). The probability of the sampled air to be of free tropospheric origin

$(P_{FT})$ is according to equation (1):

$$P_{FT} = \frac{\text{PDF}_{FT}(^{222}\text{Rn})}{\text{PDF}_{FT}(^{222}\text{Rn}) + \text{PDF}_{BLI}(^{222}\text{Rn})} \frac{1}{1 + e^{k(N_{90} - N_{90,\text{TH}})}} \tag{1}$$

where PDF is the probability density function with the FT or BLI fit parameters, respectively, according to:

$$\text{PDF}_{FT}(^{222}\text{Rn}) = \frac{1}{\sigma_{FT}\sqrt{2\pi}} e^{-\frac{1}{2}\left(\frac{\log_{10}\left(^{222}\text{Rn}\right) - \mu_{FT}}{\sigma_{FT}}\right)^2} \tag{2}$$

In the present study, $NO_y/CO$ was not used because $NO_y$ measurements were discontinued at the JFJ in March 2020.

Figure 4 shows an example of a SDE with all introduced tracers, the air mass type and the INP concentration. In Figure 4b, the attenuated backscatter of the ceilometer at KSE showed an increase from background levels to $\sim 1$ m$^{-1}$sr$^{-1}$ at altitudes similar to the JFJ after midnight on July 10, indicating the presence of an aerosol plume. At 11:00 UTC, the signal of the aerosol plume was attenuated by the low-level clouds during the remaining period of the plume event. $\alpha_{SSA}$ decreased to below zero with decreasing wavelength after 2:00 UTC on July 10 (Figure 4c). The signal becomes less separated with $\alpha_{SSA}$ above zero at 15:00 UTC and noisy after midnight of July 11, indicating the end of the SDE according to this tracer. $dust_{\text{CAMS}}$ mass concentrations exceeded the threshold concentrations on July 9 at 23:00 UTC, as shown in Figure 4d, peaking at 3:00 UTC of July 10 with 19.6 µg m$^{-3}$, followed by a decay, until falling below the threshold at 20:00 UTC. FLEXPART particle surface residence times in Figure 4e indicate between 3:00 UTC on July 10 and 5:00 UTC on July 11 that the air mass is expected to have had ground contact over the Saharan domain. Following all four tracers showing a signal, the SDE was classified as hcSDE. The INP concentrations show an increase from background INP concentrations on July 9 at 23:00 UTC (Figure 4a), to concentrations above 200 INP std L$^{-1}$, followed by lower concentrations coinciding with $\alpha_{SSA}$ relaxing back to values above zero for one hour. After a brief, second increase in INP concentrations, a decline to background levels at midnight of July 7 followed. No tracer shows an identical onset and decline as observed in the INP concentrations, however, $\alpha_{SSA}$ was the closest. The BLI air masses were present during the SDE, with $^{222}$Rn initially at FT levels at 4:00 UTC on July 10, followed by a rapid change to BLI levels. At the end of the SDE, $N_{90}$ indicated FT conditions, while $^{222}$Rn still pointed to BLI, resulting in the air mass being classified as BLI.

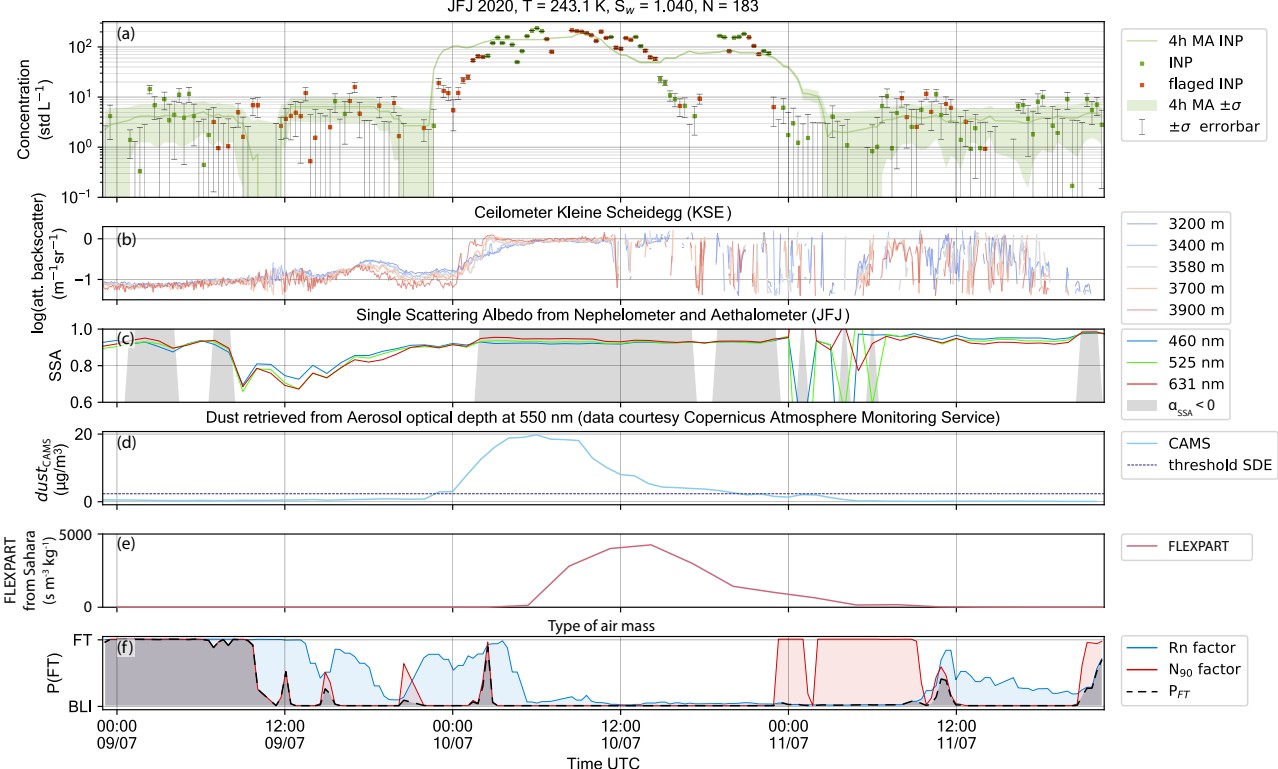

**Figure 4.** An example period classified as hcSDE between July 7 and July 11, 2020: (**a**) measured INP concentrations with HINC-Auto at $T = 243.15$ K and $S_w = 1.04$ at the JFJ. The errorbars correspond to $\pm$ 1 standard deviation and a 4-hour moving average (4h MA) is green shaded. (**b**) the attenuated backscatter signal from the ceilometer at the KSE, evaluated at different altitudes. (**c**) the single scattering albedo (SSA) and periods for which the single scattering albedo Ångström exponent is negative ($\alpha_{SSA} < 0$). (**d**) dust retrieved from satellite remote sensing generated using Copernicus Atmosphere Monitoring Service information (2020). (**e**) Cumulative surface residence time over the Saharan desert of particles arriving at the JFJ. (**f**) The probability for free tropospheric (FT) or boundary layer intruded (BLI) air mass to be sampled at the JFJ for the left hand side factor of equation (1), dependant on the $^{222}$Radon concentrations ($^{222}$Rn, blue line) and the right hand side factor, dependant on the particle concentrations with a mobility diameter larger than 90 nm ($N_{90}$, red line) as well as their product, the probability of the sampled air to be of free tropospheric origin ($P_{FT}$, black dashed line).

## 2.5 Particulate matter measurements

The mass concentration of particulate matter with an aerodynamic diameter below 2.5 μm (PM2.5) and below 10 μm (PM10) is continuously recorded with a white light optical aerosol spectrometer (Fidas 200, Palas GmbH, Germany). The data are provided with a temporal resolution of 10 min, with each data point reflecting the average particulate matter over the past 295 10 min. The spectrometer measures the intensity of single particles at an angle between 85° and 95° and infers the particle diameter solving the inverse Mie problem. The sensitivity of the spectrometer is checked 6 times per year with calibration dust.

## 2.6 CALIOP

To assess the atmospheric transport of dust and the presence and retrieved phase of nearby clouds in the case study (see subsection 3.3), post-processed lidar data from the Cloud-Aerosol Lidar with Orthogonal Polarization (CALIOP) instruments of the Cloud-Aerosol Lidar and Infrared Pathfinder Satellite Observations (CALIPSO) satellites was used. CALIPSO is part of the NASA A-train constellation (Ste, 2002), which was launched in 2006 to a sun-synchronous polar orbit with 98.2° inclination at an altitude of 705 km (Win, 2009). The data was accessed via https://www-calipso.larc.nasa.gov/products/lidar/browse_images/exp_index.php?d=2020 (last access: 10 April 2021).

## 3 Results

In the investigated time period between February 7 and December 31, 2020, 26 SDEs were detected (based on the classification described in Section 2.2), consisting of 14 hcSDEs and 12 lcSDEs (see Table A1 in the Appendix for more details), with a total duration of 55 days 20 hours, which corresponds to 17 % of the overall investigated time period. The median INP concentrations of the individual event medians was $15.3 \pm 1.2$ INP std $L^{-1}$. Analogously, the medians of all single event 25 %, and 75 % quartile INP concentrations were $9.1 \pm 1.1$, and $32.4 \pm 1.4$ INP std $L^{-1}$ (event-based; see Table 1, and Table A1 for more detail), generally one order of magnitude higher than the median INP concentration of $1.1 \pm 1.0$ INP std $L^{-1}$ during periods without SDEs (non-SDE). All SDE quartiles except the 25 % quartile exceed the $95^{th}$ percentile during non-SDE conditions. The median of all INP concentrations during all SDEs combined increases to $22.5 \pm 1.4$ INP std $L^{-1}$, rendering the longer dust events generally to contain more ice-active particles. This median concentration is consistent with previously reported values at the JFJ of 26.1 INP std $L^{-1}$ (Lacher et al., 2018a). The $75^{th}$ and $95^{th}$ percentile concentration during all SDE periods were $68.7 \pm 1.5$ and $308.1 \pm 1.4$ INP std $L^{-1}$, respectively, which is three times lower than what Lacher et al. (2018a) observed at the JFJ, and the highest concentrations measured were half as high as those measured in the Saharan Air Layer in Tenerife (1274 vs 2500 INP std $L^{-1}$ in Boose et al., 2016b). Given the proximity of Tenerife to the Sahara compared to the JFJ, the difference in peak INP concentrations is to be expected. None of the SDEs was, at least partially, without BLI. In former studies, SDEs were reported to be occurring in the FT only (Lacher et al., 2018a); however, our results indicate that FT conditions ($P_{FT} \geq$ 50 %) made up 14.5 % of the total SDE-time, compared to non-SDE periods, where FT conditions prevailed for 40.5 % of the time. A smaller FT fraction during SDE periods versus non-SDE periods is expected because of the seasonality of SDEs, with few events in winter, when also FT conditions prevail. The observed INP concentrations during SDEs were lower in FT conditions compared to periods with BLI with median concentrations of $17.3 \pm 1.1$ and $23.7 \pm 1.5$ INP std $L^{-1}$, respectively, however, they did not significantly differ, as the median INP concentrations of one class does not exceed the interquartile range INP concentrations of the other class and vice-versa. Noteworthy are the reported negative INP concentrations due to the low signal-to-noise ratio for the $25^{th}$ percentile of the overall non-SDE periods. With an uncertainty of $\pm 1.0$ INP std $L^{-1}$, the $25^{th}$ percentile is within the instrument's noise and, therefore, can statistically not be distinguished from 0 INP std $L^{-1}$ with the equipment used.

**Table 1.** Statistics of the analyzed time period with the $25^{th}$ ($Q_{25\%}$(INP)), $50^{th}$ (median), $75^{th}$ ($Q_{75\%}$(INP)) and $95^{th}$ percentile ($Q_{95\%}$(INP)), the median $dust_{CAMS}$ concentration, and the fraction of all measurements in the FT ($P_{FT}$). "All SDE periods" assesses the collective INP measurements during all SDEs, "Individual SDEs (event-based)" analyzes every single SDE and shows, e.g., the median INP concentration of all single SDE medians. All INP concentrations are in units of INP std $L^{-1}$. Uncertainty indicates a 1 $\sigma$ counting uncertainty of HINC-Auto.

| INP concentration (std $L^{-1}$) | $Q_{25\%}$(INP) | Median(INP) | $Q_{75\%}$(INP) | $Q_{95\%}$(INP) | Median($dust_{CAMS}$) (µg m$^{-3}$) | $P_{FT}$ |
|---|---|---|---|---|---|---|
| All SDE periods: | 7.4 ± 1.1 | 22.5 ± 1.4 | 68.7 ± 1.5 | 308.1 ± 1.4 | 3.73 | 14.5 % |
| Individual SDEs (event-based): | 9.1 ± 1.1 | 15.3 ± 1.2 | 32.4 ± 1.4 | 60.8 ± 1.6 | - | - |
| SDE in FT | 7.4 ± 1.1 | 17.3 ± 1.1 | 46.1 ± 1.3 | 112.7 ± 1.3 | 3.12 | 100.0 % |
| SDE with BLI | 7.4 ± 1.2 | 23.7 ± 1.5 | 74.6 ± 1.5 | 354.9 ± 1.5 | 3.86 | 0.0% |
| Overall non-SDE: | -0.9 ± 1.0 | 1.1 ± 1.0 | 3.8 ± 1.2 | 12.5 ± 1.6 | 0.15 | 40.5 % |

Southwesterly wind directions occurred relatively more frequently during SDEs than during non-SDE periods (see Figure A1 for more details). However, if the wind direction during SDE periods is compared to non-SDE periods of identical duration a week prior to each SDE, they are comparable, postulating a seasonal feature rather than a connection to the dust events.

### 3.1 Influence of SDEs on INP concentrations

Figure 5 shows box plots of the measured INP concentrations for all of the 26 SDEs, classified either as hcSDE or as lcSDE, compared to the box plot of measured INP concentrations during all non-SDE periods. Every SDE shows elevated levels of INP concentrations compared to the non-SDE periods, rendering every SDE to carry a higher INP loading than what is present during non-SDE periods. However, the concentration of INPs varies across SDEs, as the SDE with the lowest INP concentration (median = 1.7 INP std $L^{-1}$) is close to the median of background INP concentrations, and two orders of magnitude smaller than the median of the SDE with the highest INP concentration (median = 161 INP std $L^{-1}$). Three SDEs (12 % of all SDEs) showed median INP concentrations above 100 INP std $L^{-1}$.

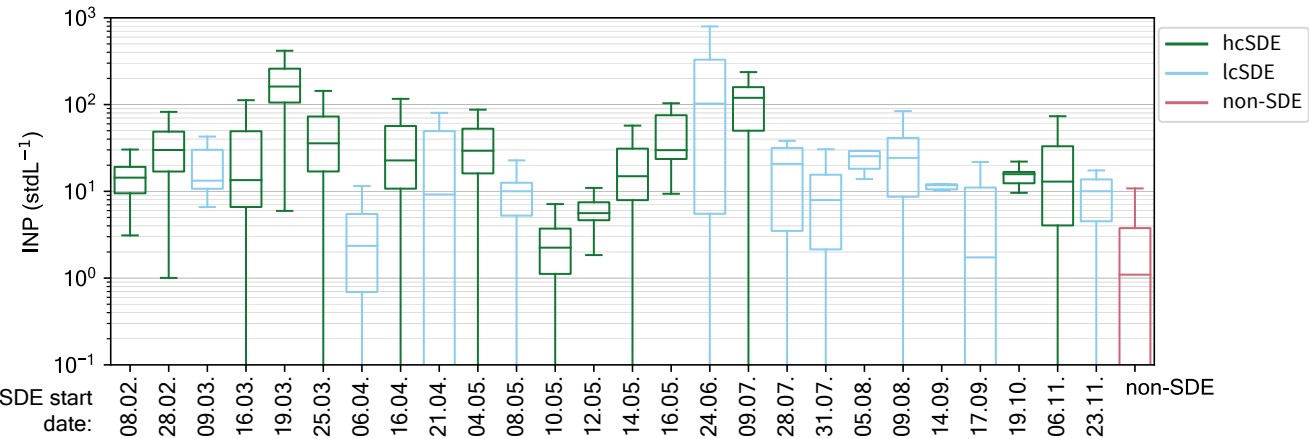

**Figure 5.** Box plots from all 26 SDEs, color-coded as high confidence SDEs (hcSDE) or low confidence SDEs (lcSDE), respectively, and the non-SDE periods: median with $25^{th}$ to $75^{th}$ percentiles, whiskers: $5^{th}$ to $95^{th}$ percentiles.

The 12 lcSDEs accounted for 33 % of the total SDE duration and showed a lower median INP concentration of 13.1 INP std $L^{-1}$ compared to 28.5 INP std $L^{-1}$ for the 14 hcSDEs. This reflects that lcSDEs have the tendency of showing lower INP concentrations than hcSDEs, yet the SDE with the lowest $95^{th}$ percentile of 7.1 INP std $L^{-1}$ was a hcSDE in May, while the SDE with the highest $95^{th}$ percentile of 881.1 INP std $L^{-1}$ was an lcSDE in June. Therefore, increased INP concentrations can be expected if at least one of the mentioned four tracers shows a SDE signal. Furthermore, all tracers showing a signal is

not indicative for highest INP concentrations to be expected. As the $\alpha_{SSA}$ is the tracer with least correspondence to the other SDE tracers, this is specially relevant for sites where the presence of dust events is inferred from the $\alpha_{SSA}$ alone, as has been done in previous studies at the JFJ (Chou et al., 2011; Boose et al., 2016a; Lacher et al., 2018a).

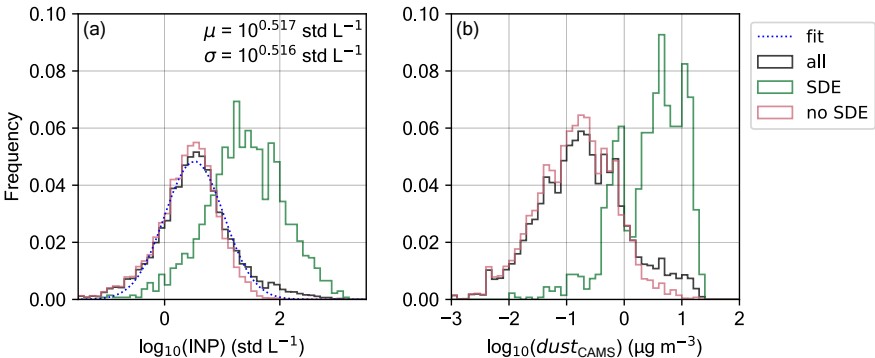

**Figure 6.** Frequency distributions of INP concentrations (**a**); and $dust_{CAMS}$ (**b**) between February 7 and December 31, 2020 (solid black), for all classified SDEs (lcSDE and hcSDE, green) and for periods without SDEs (pink). A log-normal curve with stated curve parameter in (**a**) has been fitted to the frequency distribution of all INP concentrations (dashed blue). The area under each frequency distribution is normalized to unity, which does not allow the sum of the areas below the SDE and non-SDE frequency distributions to be equal to the area below the frequency distribution of all classified SDEs.

Figure 6a shows the histogram of observed INP concentrations. The observed concentrations during SDEs are responsible for all INP concentrations above $10^2$ INP std L$^{-1}$. If a log-normal curve is fitted to all INP concentrations (blue dashed line in Figure 6a; fit parameter: $\mu = 10^{0.517}$ INP std L$^{-1}$ = 3.3 INP std L$^{-1}$ and $\sigma = 10^{0.516}$ INP std L$^{-1}$ = 3.3 INP std L$^{-1}$), both tails of the measurements (black line) are symmetric and above the fitted curve. Therefore, the frequency distribution of all INP concentrations shows a leptokurtic log-normal distribution, which is consistent with the theory by Ott (1990) and other studies (e.g., Welti et al., 2018; Schrod et al., 2020). If the fitted curve is compared to the frequency distribution of all non-SDE INP concentrations (pink line), the non-SDE INP concentrations show a distinctly less symmetric log-normal distribution, as the high concentration tail above 10 INP std L$^{-1}$ falls below the fitted curve, while the low concentration tail remains above the fitted curve. This could mean that the kind of INPs detected during SDE also contributes to the overall INP population during non-SDE periods, but chemical analysis would be necessary to categorically conclude this. After all, SDEs are operationally defined periods, where dust concentrations become so dominant that their signature can be more clearly measured and the properties of dust particles outweigh competing effects from other aerosol species (e.g., absorption, back- and forward scattering characteristics).

## 3.2 The contribution of dust to the INP concentration at 243 K

HINC-Auto samples an approximately constant volume flow of ambient air, and counts the number of INPs after exposure to the set temperature and supersaturated saturation ratio within the chamber. The ratio of INPs detected during all SDEs to all

INPs detected during the entire analysed period is calculated to quantify the contribution of Saharan dust, including a 1 $\sigma$ limit of detection (LOD) to express the uncertainty involved:

$$\frac{\sum \text{INP during SDEs} \pm \text{LOD}}{\sum \text{total INP} \pm \text{LOD}} = \frac{90938 \pm 90.4 \text{ INP}}{121691 \pm 265.1 \text{ INP}} = 74.7 \pm 0.2 \text{ \%} \tag{3}$$

Thus, 74.7±0.2 % of all INPs were detected during SDEs. Whether this is because of the ice nucleation activity of the mineral dust or because of biogenic material content, as proposed by other studies (e.g., see O'Sullivan et al., 2016; Augustin-Bauditz et al., 2016), is outside of the scope of this study. Biogenic material on the dust particles as the predominant cause of the ice nucleation activity of the particles would have two main implications. Firstly, the underlying mechanism leading to ice nucleation might differ, as proteins and other macromolecules could induce the ice nucleation as compared to topological mineral surface features, such as cracks and pores. Secondly, it would raise the question of the source of the dust particles containing ice-active biogenic material to be potentially from dried lake beds in desert regions or agricultural regions that are not differentiated in this work. In addition, a contribution of arable dust cannot be ruled out, however, during the 26 SDEs the modelled FLEXPART particles surface residence times indicated in 24 cases that the air mass had robust surface contact in the Saharan Desert, and in the other two cases, they had weak surface contact. The median INP concentration during SDE periods with FT air masses is 17.3 INP std $L^{-1}$, while with BLI it is 23.7 INP std $L^{-1}$. If we assume arable dust to show a substantially larger signal within the well-mixed PBL than in the FT, we could attribute the difference between median INP concentrations in FT SDE periods and BLI SDE periods to be because of arable dust, which is 6.4 INP std $L^{-1}$. There were three SDEs detected with lower concentrations than 6.4 INP std $L^{-1}$, one with a signal in $\alpha_{SSA}$, one without and one where the nephelometer was offline. If during SDE periods the median contribution of arable dust was 6.4 INP std $L^{-1}$, then this should be also similar during non-SDE periods. However, the median INP concentration during non-SDE periods was only 1.1 INP std $L^{-1}$. We do not see any indications why during SDEs the contribution of arable dust should be substantially larger than during non-SDE periods. If at all, we expect arable dust would contribute to BLI INP concentrations during non-SDE periods. It is fair to assume for background INPs present during non-SDE to be also present during SDE. Assuming a constant concentration over time, background INPs would contribute as little as 0.5 % to the INP population during SDEs.

Since SDEs consist only of the high concentration tail of the log-normal distribution of dust in the atmosphere after successive random dilution by large and small scale dynamics, also visible in the $dust_{\text{CAMS}}$ frequency distribution in Figure 6b, it is likely that ice-active dust is also present outside of SDE periods, however in smaller concentrations. For the following assessment, we assume that a constant mass-fraction of all dust carried to the JFJ will act as INPs. This is a bold simplification, e.g., Boose et al. (2016b) found INP concentrations to vary within a factor of 7 for the same dust mass concentration; however, it provides a rough estimate of the contribution of dust to the INP concentration at $T = 243$ K and $S_{\text{w}} = 1.04$ at the JFJ. $dust_{\text{CAMS}}$ will be used as a proxy for the ambient dust mass concentration. The ratio of $dust_{\text{CAMS}}$ advected to the JFJ during all SDEs to all advected dust to the JFJ during the entire analysed period is:

$$\frac{\sum dust_{\text{CAMS}} \text{ during SDEs}}{\sum \text{total } dust_{\text{CAMS}}} = \frac{9538 \text{ µg}}{12466 \text{ µg}} = 76.5 \text{ \%} \tag{4}$$

A literature search did not yield information about the uncertainty of $dust_{\text{CAMS}}$. If 76.5 % of the dust is responsible for 74.2±0.2 % of the INPs (74.7±0.2 % - 0.5 %), and 23.5 % of all $dust_{\text{CAMS}}$ was advected to the JFJ during non-SDE periods, , we estimate that about 23 % of the INPs measured during non-SDE periods were dust-related with our assumption that a constant mass fraction of dust acts as INPs. Therefore, the total contribution of dust to the INP population measured at the JFJ at $T$ = 243 K and saturation ratio of $S_{\text{w}}$ = 1.04 estimated to be 74 % + 23 % ≈ 97 %. Note, during non-SDE periods, dust contributed 23 / 25.3 % ≈ 91 % to the overall INP population. To validate the stated contribution in future studies or investigate the presence of biogenic material causing the ice-activity, we propose to separate INPs from the bulk aerosol population to analyze the chemical composition of the INPs as well as study the surface using scanning electron microscopy. However, to our knowledge, such equipment to separate INPs has not been used in a continuous annual study. Detailed suggestions of how this can be achieved in a long-term automated study by modifying HINC-Auto is presented elsewhere (Brunner, 2021), and is beyond the scope of the current manuscript.

We would like to emphasize that there is substantial uncertainty involved in the stated fraction, as the assumption of a constant ice-active mass-fraction in all dust particles changes with dust type (e.g., Hoose and Möhler, 2012, and references therein). Also, the uncertainty of $dust_{\text{CAMS}}$ at low concentrations is expected to be significant, but remains unquantified. Furthermore, the statement is only applicable for the stated location, temperature of $T$ = 243 K and saturation ratio of $S_{\text{w}}$ = 1.04. At warmer temperatures mineral dust shows a lower ice activity (e.g., Eastwood et al., 2008, and references therein) and subsequently plays a lesser role at the JFJ such that other species become the dominant population. This is consistent with observations of Conen et al. (2015), where INPs active at 265 K showed a correlation with ambient temperature and during SDEs were on the same order of magnitude as during non-SDE periods. This is expected, as at 265 K the dominant type of INP is different than at 243 K (e.g., biological vs. mineral dust). Also, at measurement locations further away from the dust source than the JFJ is, or locations closer to local sources in the PBL, the contribution of dust on the total INP population can be expected to be much smaller.

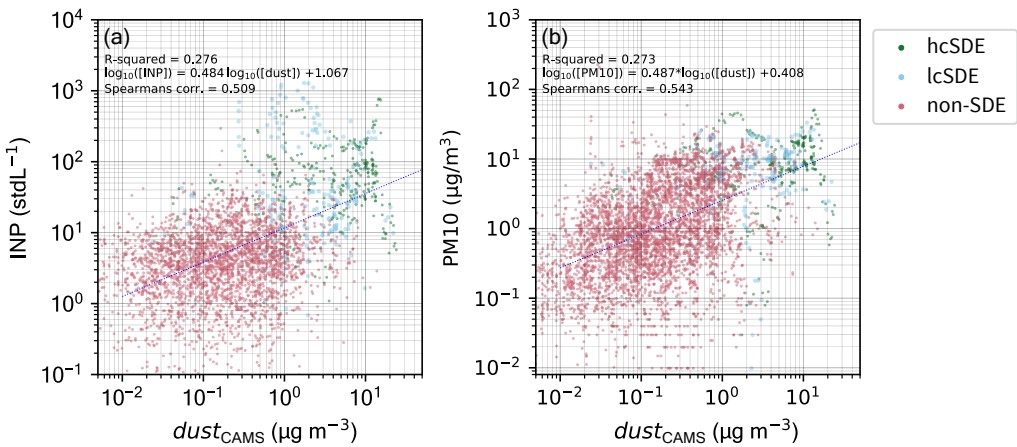

**Figure 7.** $Dust_{CAMS}$ mass concentrations with simultaneously measured INP number concentrations (**a**), and $dust_{CAMS}$ mass concentrations with simultaneously measured PM10 concentrations in (**b**), each for periods of high confidence Saharan dust events (hcSDEs), low confidence Saharan dust events (lcSDEs) and non-SDEs, which the corresponding correlation parameters and linear the fit (blue line). For either plot, the $dust_{CAMS}$ concentrations were linearly interpolated to the sampling frequency of the INP measurements (20 min) and PM10 measurements (10 min), respectively.

This raises the question whether the INP concentrations at the JFJ can be inferred meaningfully from $dust_{CAMS}$. Figure 7 shows the $dust_{CAMS}$ mass concentration at the time of every INP measurement with the corresponding INP concentrations
and PM10 concentrations. The Spearman's rank correlation coefficient of $\rho = 0.509$ indicates there is a correlation between the $dust_{CAMS}$ and INP concentrations, however the $R^2 = 0.276$ indicates there is only a weak linear correlation. This does not come as a surprise, as the INP concentration is a particle number concentration per volume of air that tends to scale with particle surface area ($\propto r^2$) for an identical INP type or air mass dominated by a certain INP species, or with the number concentration of viable dust particles, while $dust_{CAMS}$ provides a mass concentration ($\propto r^3$). Given the distribution of the
dust particles during the SDEs is not log-normal (see Figure 6b) suggests that the size of dust particles varies, and thus, the number concentrations will not scale with dust mass. Therefore, $dust_{CAMS}$ is also compared to PM10, which both are in units of mass per volume of air. Interestingly, with $\rho = 0.543$ and $R^2 = 0.273$ the pattern does not change much, indicating the level of uncertainty within $dust_{CAMS}$. A comparison of the observed PM10 with the PM10 from CAMS showed a similar agreement with $\rho = 0.491$ and $R^2 = 0.265$, and better agreement with $\rho = 0.591$ and $R^2 = 0.336$ if only concentrations above 1 µg m$^{-3}$
are considered. As the relative measurement uncertainty in lower INP concentrations and presumably also in low $dust_{CAMS}$ concentrations increases, a higher degree of correlation would be expected at high concentrations, which is, however, not the case. This analysis strengthens the caveats of our earlier conclusion regarding the dust contribution to INP concentrations at the JFJ.

## 3.3 Backtracking SDEs with the European ceilometer network

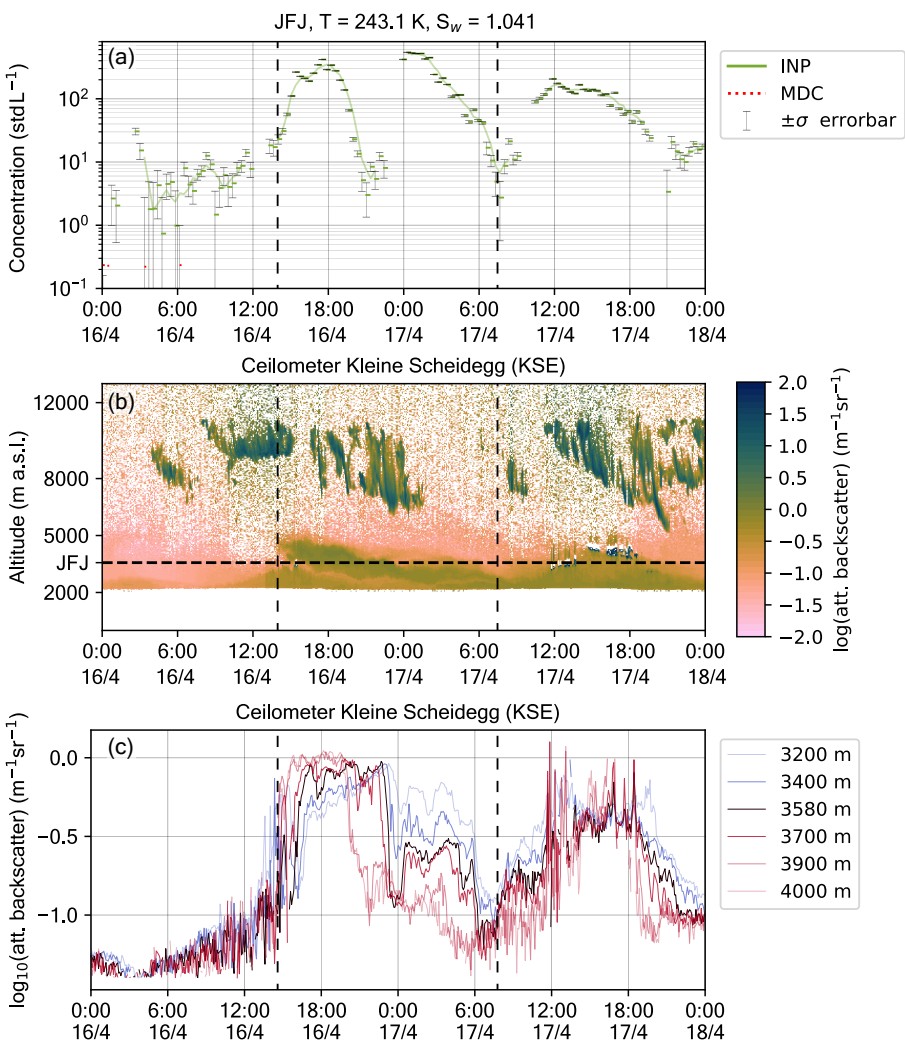

**Figure 8.** (**a**) INP concentrations and (**b**) vertical profile of the attenuated backscatter from the ceilometer at the KSE and (**c**) at selected altitudes similar to the altitude of the JFJ with filtering for in-cloud and in-precipitation signals during a classified dust event, starting on April 16, at 11:26 UTC and ending on April 17, at 22:21 UTC. When an INP measurement is below the minimum detectable concentration (MDC) of 0.236 INP std L$^{-1}$, a red dashed marker is plotted. The horizontal black dashed line indicates the altitude of the JFJ in (**b**); Vertical black dashed lines provide guidelines to better compare the three subplots.

Figure 8a shows the INP concentrations measured during a classified SDE in spring, peaking at 419 INP std L$^{-1}$. Figure 8b shows the ceilometer attenuated backscatter signal at the KSE, with the increased attenuated backscatter due to the dust particles up to 5000 m a.s.l., and in Figure 8c the attenuated backscatter evaluated at similar altitudes as the JFJ, with filtered ceilometer measurements when a cloud was below or at the same altitude as the JFJ. Similar altitudes at the KSE rather than

the exact altitude of the JFJ were chosen as, depending on the wind direction, orographic lifting or subsidence is to be expected between the KSE and the JFJ. Furthermore, eddies forming on the leeward side of the mountain ridge can induce mixing of air masses from higher or lower altitude to the JFJ, depending on wind speed, wind direction and atmospheric stability. The similar trends in observed INP concentrations and attenuated backscatter are not just a feature of the SDE on April 16-17, but across all investigated data where the ceilometer signal is not attenuated by clouds or precipitation. Another example is provided in Figure A4, where it becomes apparent that the observed INP concentration does not always best agree with the backscatter signal retrieved at the altitude of the JFJ, but sometimes rather with the signal retrieved at a lower or higher altitude. This also changes over time as the wind situation changes throughout the day (e.g., see Ketterer et al., 2014).

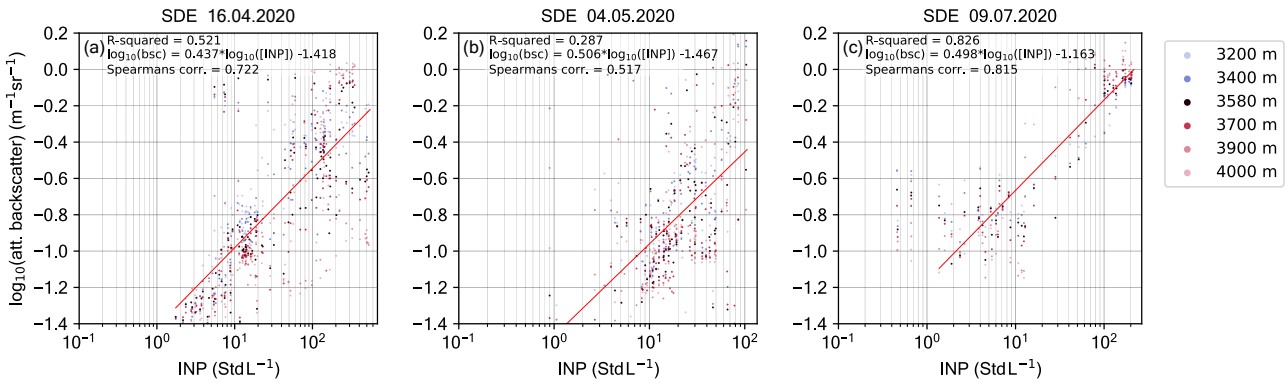

**Figure 9.** Scatter plots of simultaneous INP concentrations and attenuated backscatter signal at the KSE for a SDE starting April 16 (**a**), May 4 (**b**) and July 9 (**c**). Shown are the median ceilometer measurements during the period of each INP measurement, which the corresponding correlation parameters and the linear fit (red line).

Figure 9 shows the observed relation between the INP concentration and the attenuated backscatter of three classified dust events. With Spearman's rank correlation coefficients ranging between $0.517 \leq \rho \leq 0.815$, there is a correlation between the two quantities. As both INP concentration and the backscatter coefficient scale to a first approximation with the cumulative surface of the ambient particles, a linear fit can be applied. If a linear fit is applied to the logarithm of both quantities, including all altitude levels, the slopes of the fitted curves are very similar, but the offsets differ, resulting in a factor of up to 4.5 different INP concentrations for the same attenuated backscatter. Considering the limiting factors, like the 4.4 km distance between the measuring sites, the lifting and subsidence depending on the synoptic situation, modified by the orography and the analysis of the attenuated backscatter signal rather than the corrected backscatter signal, we consider the agreement such that it allows to use the signals of the ceilometer network to study the spatial temporal evolution of INP concentrations and investigate their atmospheric pathways. For this purpose, a fit was computed for all data (SDE + non-SDE) where the ceilometer signal is not attenuated by clouds or precipitation ($\rho = 0.63$):

$$\text{INP}_{243 \text{ K}} \text{ conc.} = 10^{2.062 \log_{10}(ABSc)+2.81} \tag{5}$$

Where INP$_{243\ K}$ conc. is the INP concentration at $T = 243$ K and $S_\mathrm{w} = 1.04$ in INP std L$^{-1}$ and $ABSc$ is the attenuated backscatter in m$^{-1}$ sr$^{-1}$. We propose the fit in future work to be validated at other locations where ceilometer data are available. As previously shown, the INP concentrations at the JFJ are dominated by mineral dust particles, which renders the obtained fit likely to only be valid for when mineral dust is the dominant species amongst the INP population.

The ceilometer at the KSE often showed upstream virga prior to dust plumes arriving at the JFJ. These virga are starting at an altitude above 5000 m a.s.l. and appear to be connected to dust plumes (see Figure 8b). To illustrate this, two dust events were analyzed in more detail. Figure 10a shows a map of the dust concentration from CAMS on February 9, 2020, 6:00 UTC with the location of three ceilometer stations Freiburg (FRE) in Southern Germany, KSE and St. Auban (StA) in the South of France. An anticyclone with a surface pressure of 1030 hPa at sea-level centered over Serbia extended to the Alps, while a weak cold front with core above Iceland passed Germany. Figure 10b correspondingly shows the CAMS dust concentration on April 16, 2020, 6:00 UTC, with the ceilometer stations Deuselbach (DEU) in the Western part of Germany, and the KSE, where an anticyclone with a center over Croatia extended to central France and Germany with a surface pressure of 1020 hPa at sea-level.

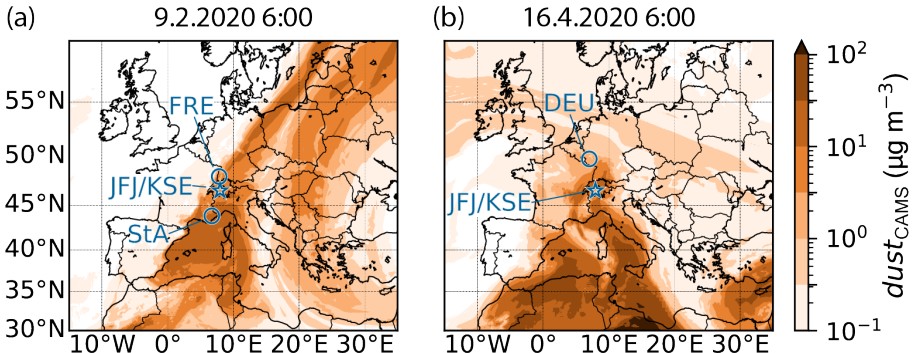

**Figure 10.** Snapshot of the spatial distribution of $dust_{\mathrm{CAMS}}$ during two dust events, (**a**) for February 9, 6:00 UTC, and (**b**) for April 16, 6:00 UTC, with the locations of the ceilometers Freiburg (FRE), Kleine Scheidegg (KSE), St. Auban (StA), and Deuselbach (DEU).

Figure 11 shows the ceilometers StA, KSE, and FRE between February 7 - 10. On February 8, at 23:29 UTC, INP concentrations exceeded background concentrations, indicating the onset of a SDE, and peaked on February 9 at 2:59 UTC with 42 INP std L$^{-1}$. The corresponding signal can be seen in the attenuated backscatter at the KSE during the same period, initially at an altitude between 3500 and 4600 m a.s.l. and descending over time. A faint signature of the plume is visible before February 8, 23:30 UTC at the altitude range between 4500 and 5500 m a.s.l. and going back until February 8, 13:00 UTC, where it connects to signatures from sedimenting hydrometeors with equal upper and lower limits. These hydrometeors in turn appear to have sedimented from 11400 m a.s.l., forming a virga. Analogous behaviours in the signals of the plume with each a corresponding virga are also visible in the ceilometers StA and FRE (see Figure 11), however, the signature of the plume is more distinct. Radio soundings at Payerne, 84 km northwest of the KSE, observed on February 8, at 0:00 UTC at 11400 m a.s.l. a temperature of 217 K and saturation ratio with respect to liquid water and ice of $S_\mathrm{w} = 0.39$ and $S_\mathrm{i} = 0.68$, respectively,

well below the saturation ratio needed for the homogeneous nucleation of solution droplets of $\sim$ $S_i = 1.52$ at 217 K (Koop et al., 2000). The tropopause was above 11650 m a.s.l. according to radio soundings. Ice saturation was detected between 8850 and 8880 m a.s.l., and between 10200 and 11200 m a.s.l.. Below 7790 m a.s.l., the saturation dropped from $S_i \geq 0.8$ to $S_i \leq$ 0.15. This raises the hypothesis, whether the ice-active particles nucleated the ice clouds at altitudes above 7790 m a.s.l., with subsequent sedimentation of the ice crystals to below 7790 m a.s.l., where they sublimated leaving behind dry INPs. These INPs further sedimented, however, due to the lower mass, at a much smaller rate. Thus, signals of the plume appear in the ceilometer more elongated after sublimation (e.g., in FRE at 13:00 UTC on February 8) compared to prior in the virga. What opposes this hypothesis is the fact, that no Saharan dust plume can be observed in the ceilometer measurements at altitudes above 7790 m a.s.l.

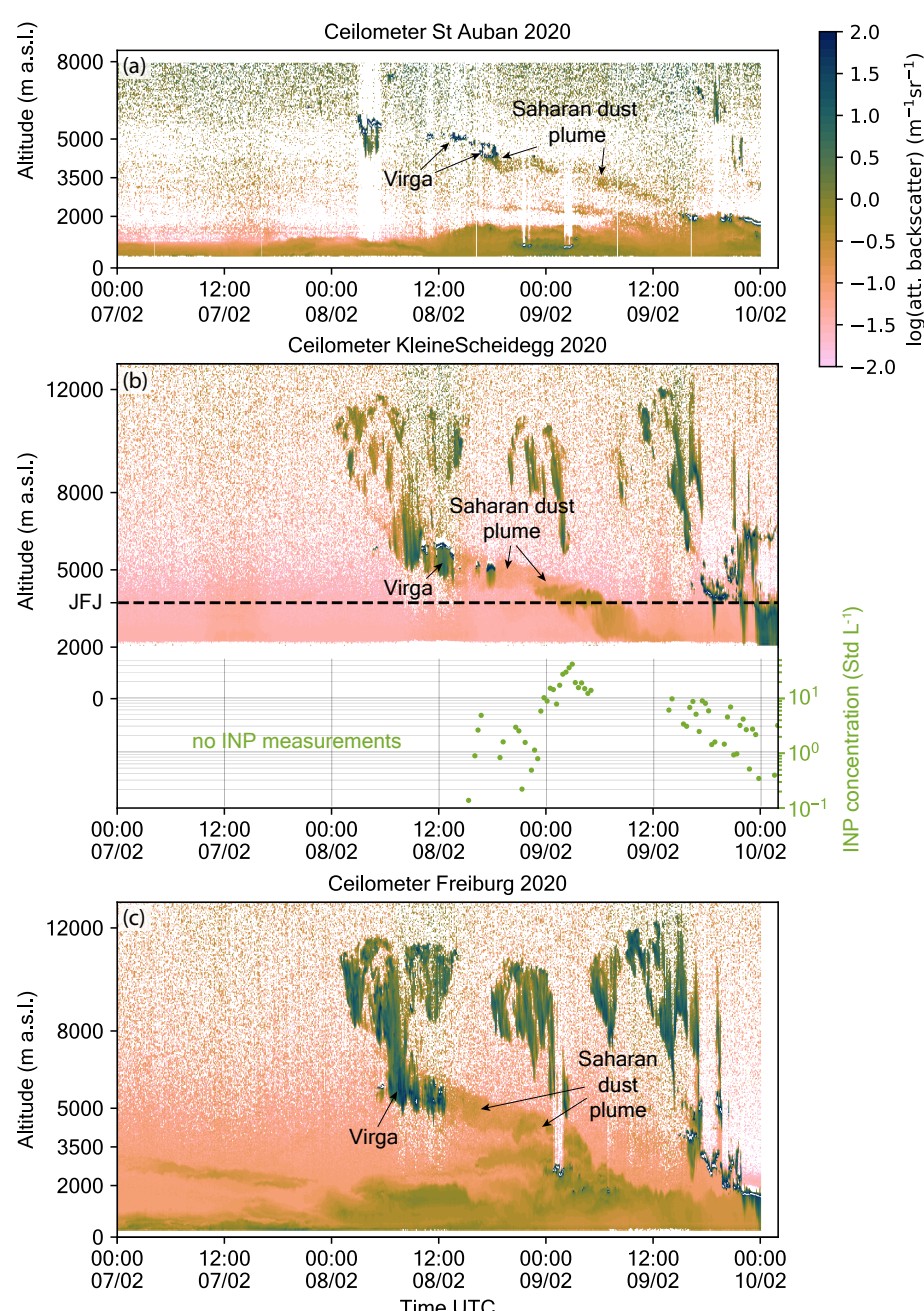

**Figure 11.** Vertical profiles of the attenuated backscatter from the ceilometer at St. Auban (StA, **a**), Kleine Scheidegg (KSE) with INP concentrations at the JFJ (**b**), and Freiburg (FRE, **c**), with the dust plume and connecting virga visible between February 8, 3:00 UTC and February 9, 12:00 UTC.

Figure 12 shows another example of the observed pattern , with the ceilometers KSE and DEU. The simultaneous ceilometer and INP measurements at the KSE and at the JFJ, respectively, which are shown in Figure 12a, indicate that the Saharan dust

plume on April 16-17 contains INP number concentrations above 100 INP std $L^{-1}$, however, no connection between the Saharan dust plume and a virga is apparent. The Saharan dust plume can be tracked north to other ceilometer locations, such as DEU, following the outline of the dust plume in Figure 10b. At DEU, shown in Figure 12b, the lower end of the virga and the onset of the dust plume are collocated as before in the case study from February 8. However, also in this case no signal of the Saharan dust plume is apparent above 5500 m a.s.l. The sounding at Idar-Oberstein from the Deutscher Wetterdienst (15 km East of DEU) reported a tropopause at 10200 m a.s.l. and ice supersaturation between 8370 and 10370 m a.s.l.. Below 8370 m a.s.l., there was a steady decrease from ice saturation to $S_i = 0.17$ at 5100 m a.s.l.. A three-dimensional kinematic backward trajectories analysis was carried out using Lagranto (Wernli and Davies, 1997), with wind fields from the European Centre for Medium-Range Weather Forecasts (ECMWF) Integrated Forecasting System (IFS) HRES model with a horizontal resolution of $0.1° \times 0.1°$. The analysis showed trajectories to originate from the Saharan desert within Algeria, where they were close to the surface ($\leq 300$ m above ground) on April 14, followed by an ascent to 300 hPha ($\sim 9000$ m a.s.l.) above the KSE. Observations from the CALIOP lidar onboard the Cloud-Aerosol Lidar and Infrared Pathfinder Satellite (CALIPSO) show large amounts of dust over Algeria up to 6000 m a.s.l. on April 14. On April 15, CALIPSO did not pass over the area west of Sardinia, where the backward trajectories would predict the location of the air mass containing the Saharan dust. On April 16 at 12:30 UTC, CALIOP retrieved an aerosol signature between 11000 to 13000 m a.s.l. over central Italy (43.3° N, 12.7° E), surrounded by cirrus clouds, but could not identify the type of aerosol. Assessing all presented data, the hypothesized pathway of mineral dust being lifted to cirrus altitudes, where it nucleates ice to form cirrus clouds, sediments within the ice crystals to lower altitudes, where after ice-sublimation the dust is left as residuals, is plausible, yet remains to be proven at this point. We plan on addressing the hypothesis in future work. Such a transport pathway poses some relevant implications for atmospheric processes. First, simulated back-trajectory analysis will not well reproduce the history, and thus, the origin of the particles, as models back-track air parcels and not particles, which are subject to varying settling velocities when ice is nucleating and growing on them. And secondly, after sublimation of the surrounding ice, the residual mineral dust particles can be pre-activated, e.g., by retaining pore ice due to the inverse Kelvin effect (Marcolli, 2014). If the pre-activated particles are again exposed to ice supersaturated conditions, spontaneous ice crystal growth can occur at much lower $S_i$ than with other INPs (David et al., 2019). To capture and study this effect, design adaptations need to be implemented in our sampling equipment in upcoming studies, as the pore ice sublimates or melts in the heated and dried sampling lines upstream of the INP counter used here. We attempted such a measurement of the number concentration of pre-activated particles in this way, using a supercooled diffusion dryer at the JFJ before directly sampling the ambient particles with HINC, in addition to sampling with a reference chamber with the standard heated inlet. Nevertheless, we were unable to reach any reliable conclusions. In part, due to the clogging of the inlet from frost build-up, the conditions along the trajectory of the potentially pre-activated particles had to be precisely adjusted, such that neither particle sedimentation due to activation, nor pore ice sublimation occurs. Furthermore, the set conditions do not affect all pore sizes similarly, and pore-ice can melt if the pore ice shrinks below the critical size of an ice embryo within the pore with a constrained width. In summary, there were too many parameters that needed to be simultaneously and meticulously controlled.

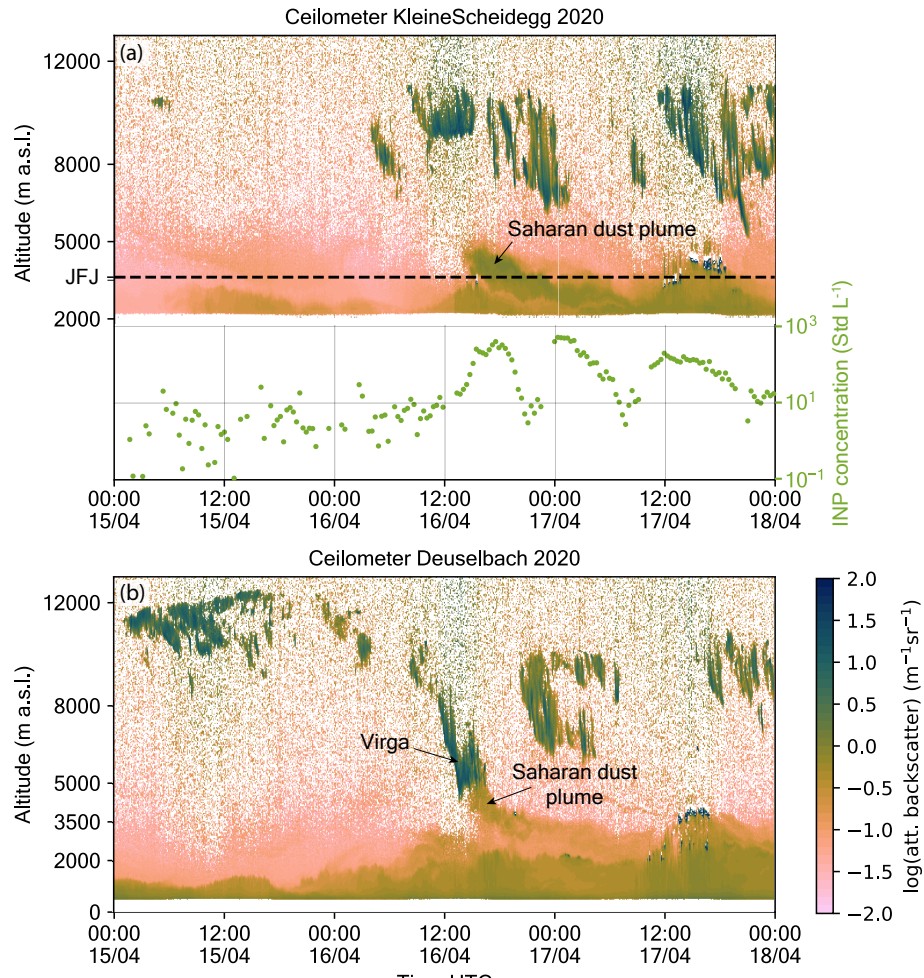

**Figure 12.** Vertical profile of the attenuated backscatter from the Kleine Scheidegg (KSE) and INP concentration at the JFJ (**a**), and vertical profile of the attenuated backscatter from Deuselbach (DEU, **b**), with the dust plume and connecting virga visible between April. 16, 6:00 UTC and 22:00 UTC.

## 4   Conclusions

We analyzed SDEs and their impact on INP number concentrations (at $T = 243.15$ K and $S_w = 1.04$) with an automated online INP counter (HINC-Auto) at the JFJ between February 7 and December 31, 2020. Twenty-six SDEs were detected, which contributed to 17 % of the total time analyzed, and showed a spread of two orders of magnitude in median INP concentrations. Every SDE showed increased INP concentrations with respect to non-SDE periods, however, not every event was significantly distinct from background INP concentrations. The median INP concentration during all SDE periods was $22.5 \pm 1.4$ INP std $L^{-1}$, which is on par with previous studies, and the $75^{th}$ and $95^{th}$ percentile were $68.7 \pm 1.5$ and $308.1 \pm 1.4$ INP std $L^{-1}$, respectively. The observed INP concentrations follow a leptokurtic log-normal distribution in accordance with theory and other

studies (e.g., Welti et al., 2018; Schrod et al., 2020). We found dust to be the main contributor to the INP population at 243.15 K and $S_w$ = 1.04 at the JFJ, with 74.7 $\pm$ 0.2 % of all INPs observed during SDEs. Based on an analysis using satellite retrieved dust mass concentrations from CAMS, we estimate that 97 % of all INPs are from dust particles or surface features on dust particles, where SDEs are just the high concentration tail of the ice-active dust particles frequency distribution. Assessing the derived surface area from independent particle size distribution measurements, we expect atmospheric INP concentrations

during SDEs are higher than reported INP concentrations, given 35 % of the total particle surface area is due to particles above the upper size cut-off of $d = 2.5$ μm of the used INP sampling equipment. Furthermore, we confirm that the attenuated backscatter signal from ceilometers in absence of cloud or precipitation can be used to study the atmospheric pathway and temporal evolution of INPs in dust plumes. We found examples of SDEs with upstream virga from altitudes above 8000 m a.s.l., which led to the hypothesis of INPs being transported to the midlatitudes, where they nucleate ice at altitudes above 5500

m a.s.l. and sediment to lower altitudes where they sublimate in drier air, and act as INPs at these lower altitudes. This could have important implications, as these INPs can be pre-activated and/or were subjected to atmospheric processing during the freeze-thawing cycles. This hypothesis will be subject of a future study, as pre-activated INPs lose their pore ice in the heated and dried sampling lines used in this study.

*Data availability.* The data presented in this publication will be made available at DOI: xx.xxxx/ethz. @Note by authors: Data will be made
available upon acceptance of publication

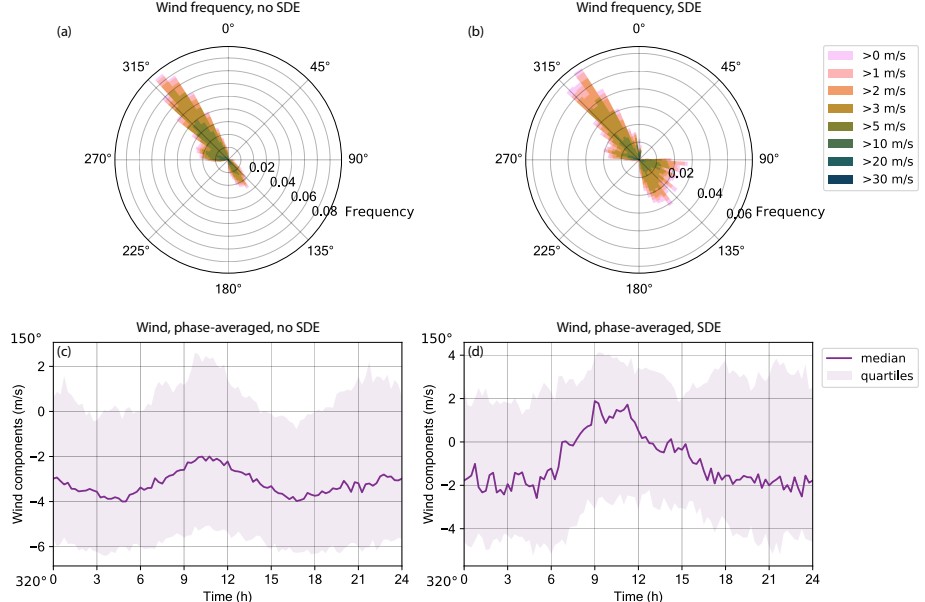

**Figure A1.** Observed wind frequency for each direction and wind velocity at the JFJ between February 7 and December 31, 2020, for non SDE periods (**a**) and for SDE periods (**b**). Furthermore, phase averaged plot of the wind components in the two principal wind directions (150° and 320°) for non SDE periods (**c**) and for SDE periods (**d**). During SDE periods, Southeasterly winds were more frequent, leading to median winds from Southeasterly directions during 7:30 and 12:30 UTC, whereas during non-SDE periods, the median wind never came from Southeasterly directions. By taking a sample of non-SDE winds in each case one week prior to a SDE, the phase average (not shown) is nearly identical to the phase averaged plot shown here for SDE periods, pointing to a seasonal feature rather than one constrained to SDEs.

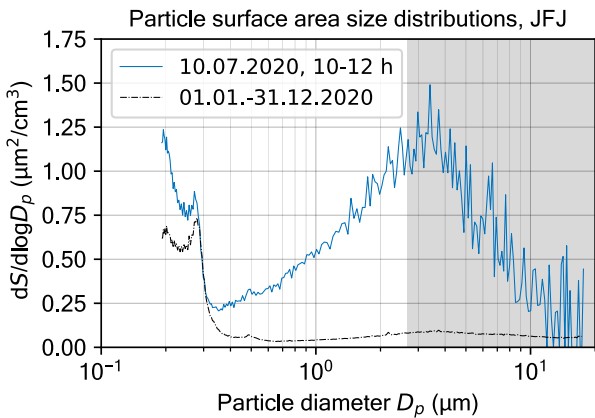

**Figure A2.** Mean particle surface area size distributions at the JFJ during a SDE at 10:00-12:00 UTC on July 10, 2020 (blue line) and between January 1, and December 31, 2020 (black dashed line), both measured with an OAS. Only particles with diameters < 2.5 µm are sampled by HINC-Auto due to the sampling line geometry and flow rates used. All particle >2.5 µm (gray shading) between February 7 and December 31, 2020 contribute during all SDE periods to 35 % to the overall particle surface area. Surface area calculations assumed perfectly spherical particles.

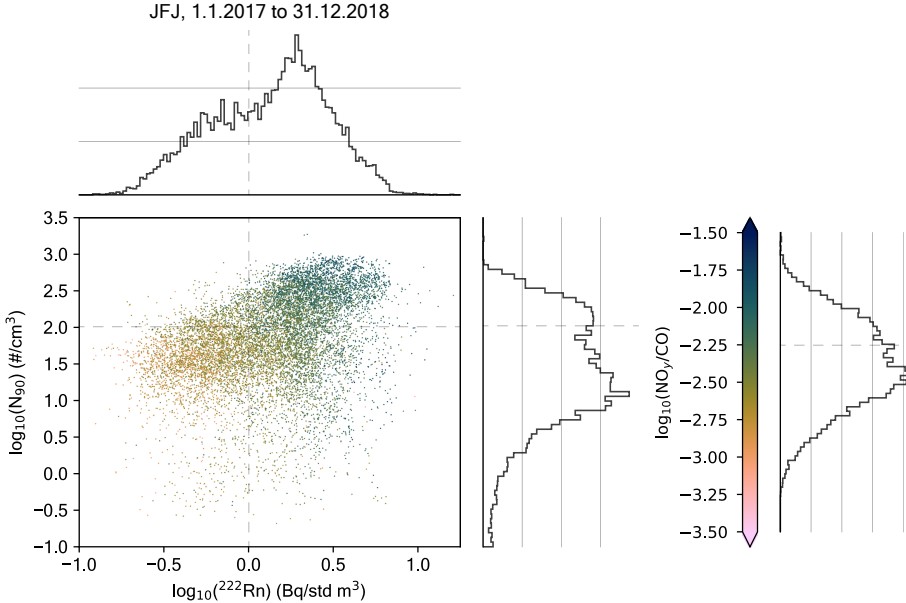

**Figure A3.** A scatter plot of simultaneous 222-Radon ($^{222}$Rn) concentrations, particle concentrations with a mobility diameter larger than 90 nm ($N_{90}$) and the ratio of reactive nitrogen species to carbon monoxide ($NO_y/CO$) as color code, between January 2017 and December 2018. These are all tracers used to detect boundary layer intrusions (BLI) at mountain top sites, where the grey dashed lines indicate the thresholds stated in literature (see Griffiths et al., 2014; Herrmann et al., 2015; Zanis et al., 2007, for more information), where low $^{222}$Rn and $N_{90}$ concentrations and low $NO_y/CO$ ratios are indicative of free tropospheric air masses (FT). We argue that $N_{90}$ concentrations above the threshold are indicative for BLI, however low $N_{90}$ concentrations don't imply FT conditions, as in the lower right quadrant both $^{222}$Rn and $NO_y/CO$ suggest BLI despite $N_{90}$ concentrations below its threshold.

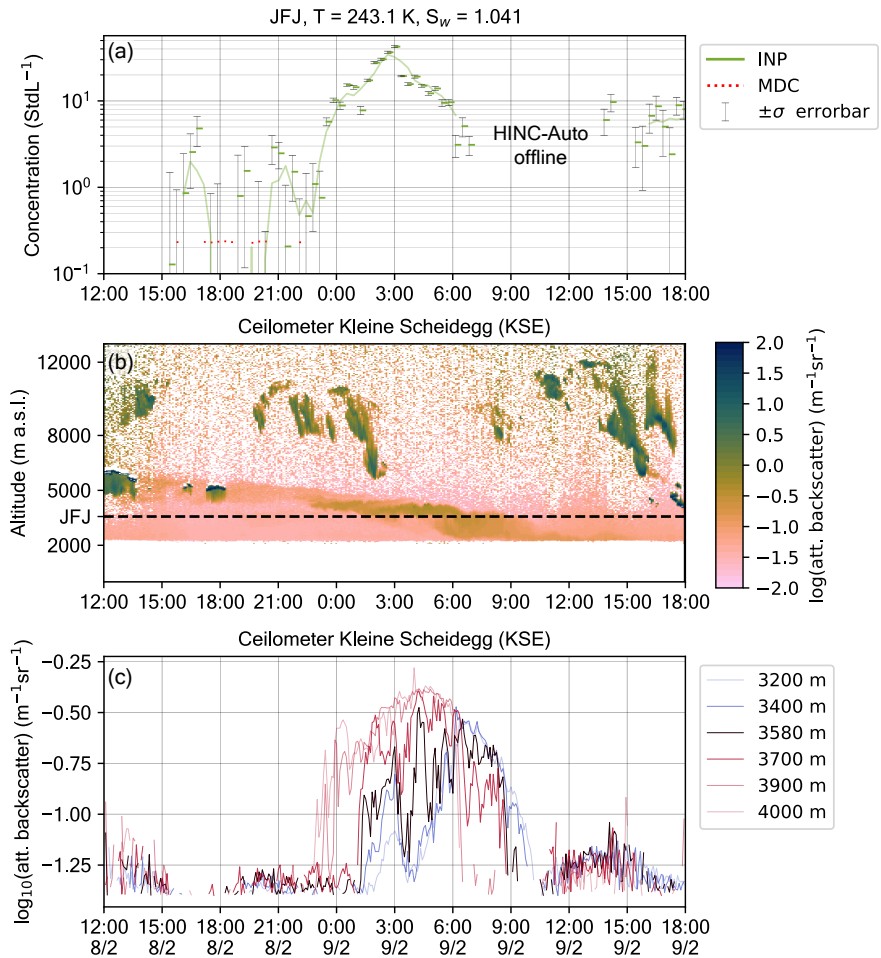

**Figure A4.** INP concentrations (**a**) and vertical profile of the attenuated backscatter from the ceilometer at the KSE (**b**) and at selected altitudes similar to the altitude of the JFJ with filtering for signals in cloud and in precipitation (**c**) during a classified dust event, starting on February 8, 23:29 UTC and ending on February 9, 6:0 UTC. When an INP measurement is below the minimum detectable concentration (MDC) of 0.236 INP std L$^{-1}$, a red dashed marker is plotted. On February 9, between 7:00 and 13:00 HINC-Auto was not sampling INPs, but different other classification and calibration measurements were performed, e.g., the particle loss rates through the sampling line as shown in the Appendix A.

Tab. A1: Detailed statistics of the analyzed time period with the $25^{th}$ ($Q_{25\%}$(INP)), $50^{th}$ (median), $75^{th}$ ($Q_{75\%}$(INP)) and $95^{th}$ percentile ($Q_{95\%}$(INP)), all as INP concentration (std L$^{-1}$), and the fraction of all measurements in the FT ($P_{\text{FT}}$). Overall SDE includes all INP measurements during SDEs, Overall SDE event-based analyzes every SDE and shows, e.g., the median $Q_{25\%}$(INP). "Y" indicates that the tracer showed a SDE signal, "N" no signal, "W" a weak signal, "C/P" that low clouds or precipitation were hiding a potential signal in the ceilometer and "-" that no data are available for the period of interest. The $dust_{\text{CAMS}}$ threshold (TH) is 2.36 µg m$^{-3}$. Dust events are classified into high confidence SDEs (hcSDEs) and low confidence SDEs (lcSDEs).

**Table A1.**

| No. | SDE start | SDE end | INP start | INP end | $Q_{25\%}$(INP) | Median(INP) | $Q_{75\%}$(INP) | $Q_{95\%}$(INP) | $\alpha SSA<0$ | Ceilo. KSE | $dust_{CAMS} <$ TH | Flexpart | $P_{FT}$ | Classif. |
|---|---|---|---|---|---|---|---|---|---|---|---|---|---|---|
| 1 | 08.02 14:00 | 09.02 06:00 | 08.02 23:29 | 09.02 06:50 | 9.5 | 14.4 | 19.1 | 36.6 | Y | Y | Y | Y | 90.5 % | hcSDE |
| 2 | 28.02 17:00 | 29.02 20:29 | 28.02 19:54 | 29.02 20:29 | 16.9 | 30.0 | 48.8 | 79.0 | Y | C/P | Y | Y | 96.7 % | hcSDE |
| 3 | 09.03 13:56 | 11.03 07:00 | 09.03 13:56 | 10.03 03:59 | 10.7 | 13.3 | 30.1 | 36.1 | Y | C/P | N | W | 73.7 % | lcSDE |
| 4 | 16.03 11:00 | 18.03 14:38 | 17.03 01:49 | 18.03 14:38 | 6.6 | 13.5 | 49.3 | 116.3 | - | - | Y | Y | 75.0 % | hcSDE |
| 5 | 19.03 16:00 | 22.03 21:10 | 20.03 23:51 | 22.03 21:10 | 105.5 | 161.4 | 259.3 | 420.9 | Y | Y | Y | Y | 4.0 % | hcSDE |
| 6 | 25.03 21:00 | 31.03 17:38 | 27.03 01:30 | 31.03 17:38 | 17.0 | 35.8 | 72.9 | 105.0 | Y | Y | Y | Y | 15.2 % | hcSDE |
| 7 | 06.04 08:00 | 07.04 12:00 | 06.04 14:00 | 07.04 04:00 | 0.7 | 2.4 | 5.5 | 9.7 | Y | Y | N | W | 94.6 % | lcSDE |
| 8 | 16.04 11:26 | 17.04 22:21 | 16.04 11:26 | 17.04 22:21 | 16.8 | 107.8 | 212.6 | 503.3 | - | Y | Y | Y | 1.5 % | hcSDE |
| 9 | 21.04 06:00 | 21.04 16:00 | 21.04 09:39 | 21.04 12:28 | -4.5 | 9.2 | 49.5 | 69.0 | W | BL | Y | Y | 0.0 % | lcSDE |
| 10 | 04.05 07:00 | 05.05 18:32 | 04.05 14:04 | 05.05 18:32 | 16.1 | 29.4 | 52.7 | 86.9 | Y | Y | Y | Y | 14.6 % | hcSDE |
| 11 | 08.05 13:00 | 10.05 10:00 | 08.05 19:19 | 09.05 12:14 | 5.3 | 10.0 | 12.6 | 16.4 | N | Y | Y | Y | 0.0 % | lcSDE |
| 12 | 10.05 13:42 | 11.05 05:12 | 10.05 13:42 | 11.05 05:12 | 1.1 | 2.2 | 3.7 | 7.1 | N | Y | Y | Y | 0.0 % | hcSDE |
| 13 | 12.05 23:39 | 13.05 15:00 | 12.05 23:39 | 13.05 03:30 | - | - | - | - | Y | C/P | Y | Y | 0.0 % | hcSDE |
| 14 | 14.05 18:00 | 17.05 05:00 | 14.05 20:48 | 15.05 23:02 | 7.9 | 14.9 | 31.1 | 52.6 | Y | Y | Y | Y | 0.0 % | hcSDE |
| 15 | 16.05 18:00 | 17.05 02:00 | 16.05 19:00 | 17.05 02:00 | 23.6 | 29.8 | 75.4 | 95.3 | Y | C/P, PAY: Y | Y | Y | 0.0 % | hcSDE |
| 16 | 24.06 02:59 | 28.06 23:08 | 24.06 02:59 | 28.06 23:08 | 5.5 | 87.5 | 329.3 | 881.1 | N | C/P, PAY: Y | Y | Y | 0.0 % | lcSDE |
| 17 | 09.07 22:59 | 10.07 22:34 | 09.07 22:59 | 10.07 22:34 | 50.0 | 119.6 | 158.8 | 209.8 | Y | C/P, PAY: Y | Y | Y | 24.1 % | hcSDE |
| 18 | 28.07 03:00 | 29.07 02:45 | 28.07 10:40 | 29.07 02:45 | 3.5 | 20.7 | 31.6 | 37.9 | N | Y | Y | Y | 0.0 % | lcSDE |
| 19 | 31.07 23:50 | 02.08 00:16 | 31.07 23:50 | 02.08 00:16 | 2.1 | 7.9 | 15.5 | 27.6 | Y | Y | Y | Y | 0.0 % | lcSDE |
| 20 | 05.08 18:00 | 08.08 12:00 | 05.08 22:55 | 08.08 00:41 | 18.2 | 25.4 | 29.2 | 52.6 | Y | W | N | Y | 100.0 % | lcSDE |
| 21 | 09.08 13:00 | 13.08 20:27 | 10.08 02:10 | 13.08 20:27 | 8.7 | 24.3 | 41.3 | 180.3 | N | Y | Y | Y | 2.6 % | lcSDE |
| 22 | 14.09 03:30 | 15.09 20:00 | 14.09 18:34 | 14.09 20:44 | 10.5 | 11.8 | 12.1 | 16.3 | N | W | Y | Y | 0.0 % | lcSDE |
| 23 | 17.09 16:00 | 20.09 23:00 | 19.09 16:45 | 20.09 14:15 | 0.1 | 1.7 | 11.1 | 18.0 | - | ? | Y | Y | 0.0 % | lcSDE |
| 24 | 19.10 09:00 | 23.10 09:00 | 21.10 19:54 | 21.10 22:04 | 12.4 | 15.8 | 16.7 | 20.4 | Y | C/P, PAY: Y | Y | Y | 0.0 % | hcSDE |
| 25 | 06.11 09:00 | 09.11 03:00 | 07.11 15:58 | 09.11 02:48 | 4.1 | 13.0 | 33.1 | 73.3 | Y | Y | Y | Y | 0.0 % | hcSDE |
| 26 | 23.11 12:00 | 25.11 03:00 | 24.11 06:49 | 24.11 07:32 | 4.5 | 10.0 | 13.7 | 16.7 | W | W | N | Y | 0.0 % | lcSDE |
| Overall SDE: | | | | | 7.4 | 22.5 | 68.7 | 277.3 | | | | | 14.5 % | |
| Overall SDE (event-based): | | | | | 9.1 | 15.3 | 32.4 | 60.8 | | | | | 0.0 % | |
| Overall non-SDE: | | | | | -0.9 | 1.1 | 3.8 | 12.5 | | | | | 40.5 % | |

**Transmission fraction of the sample line**

**Table A2.** Transmission fraction of ambient particles during a Saharan Dust event at the JFJ on February 9, 2020 over 15 min and * February 6, 2021 over 190 min.

| | 1.0 $\mu$m | 1.5 $\mu$m | 2.0 $\mu$m | 2.5 $\mu$m | 3.0 $\mu$m | 4.0 $\mu$m |
|---|---|---|---|---|---|---|
| Outside, next to Total Inlet | 100 % | 100 % | 100 % | 100 % | 100 % | 100 % |
| After Dryer + Valve | 72 % | 59 % | 40 % | - | 21 % | 14 % |
| After Chamber, $T$ = 293.15 K and $S_w$ = 0.02 | 69 % | 58 % | - | 33 % | 0 % | 0 % |
| After Chamber*, $T$ = 243.15 K and $S_i$ = 1.0* | - | - | - | - | 0.019 % | 0.017 % |

Table A2 shows transmission fraction of ambient particles trough the sampling lines at the JFJ. During a SDE on Feb 9, 2020,
differential measurements with two optical particle counters (OPC, MetOne GT-526S) were performed. One OPC measured the ambient air next to the total aerosol inlet, with the six available size bins set to $d > 1.0$ μm, $d > 1.5$ μm, $d > 2.0$ μm, $d > 2.5$ μm, $d > 3.0$ μm, and $d > 4.0$ μm. For the "After Dryer + Valve"-setup, the other OPC measured the particle concentration after the sampling diffusion dryer and 3-way valve of HINC-Auto, upstream of where normally HINC-Auto would be connected (see Brunner and Kanji (2021), Figure 2b for more details). Because the OPC was directly connected to the sample line, the
OPC's default sample flow rate of 2.83 std L min$^{-1}$ was used instead of the sample flow rate of 0.283 std L min$^{-1}$, which is used by default when HINC-Auto is connected. It can be expected that the loss rate of large particles are smaller because of the larger volume flow. The loss rates were calculated from the differences in a 15 minutes cumulative particle count measured in parallel with both OPCs. For the "After Chamber, $T$ = 293.15 K and $S_w$ = 0.02"-setup, the second OPC was connected to the outlet of HINC-Auto as it is done for INP measurements. Both chamber walls were held at a constant temperature of $T$
= 293.15 K and the filter paper within the chamber was removed, leading to no humidification of the sampled air within the chamber, and thus, keeping it at $S_w$ = 0.02 as reached after the diffusion dryers. During a SDE on Feb 6, 2021, with particulate matter with PM10 peaking at 767.8 μg m$^{-3}$, HINC-Auto was held at $T$ = 243.15 K and $S_i$ = 1.0 to test the particle survival rate without water supersaturation. The OPC was mounted as default for INP measurement at the chamber exit, but in contrast to the measurement on Feb 9, 2020, the $d > 3.0$ μm and $d > 4.0$ μm size bins in the OPC were selected. The reference particle
counts were obtained by cumulative particle concentrations over the same time period, measured by a FIDAS 200. Only 0.096 % and 0.017 % of particles $d > 3.0$ μm and $d > 4.0$ μm, respectively, penetrated the sampling line and were not lost in the sample line and were sampled within HINC-Auto.

*Author contributions.* CB wrote the manuscript with input from BB, MC, FC, SH, MH, MGB, MS, and ZAK. ZAK conceived the field study. CB conducted the INP measurements and analyzed all INP, CAMS and ceilometer data. CB interpreted the INP with ZAK, the CAMS data, the SSA with MC), FLEXPART data, ceilometer data with MH and the Radon data with FC. CB prepared the figures. BB and MGB provided the total aerosol sample line, contributed aerosol particle concentrations, $N_{90}$, absorption and scattering characteristics. MC contributed data on the SSA. FC contributed data on the Radon concentration and developed the air mass classification (FT or BLI). SH contributed FLEXPART data. MH contributed ceilometer data. MS contributed data on trace gases and PM. ZAK supervised the project and obtained funding.

*Competing interests.* The authors declare that they have no conflict of interest.

*Acknowledgements.* The authors gratefully acknowledge the Copernicus Atmosphere Monitoring Service for provding the CAMS data, the Centre for Environmental Data Analysis (CEDA) for providing the ceilometer data, and the Swiss contribution to ICOS (https://www.icos-ri.eu) for financially supporting the operation of the radon detector. The CALIPSO data were generously provided by the NASA Langley Research Centre Atmospheric Science Data Centre (https://www-calipso.larc.nasa.gov/products/lidar/browse_images/exp_index.php?d=2020). We thank MeteoSwiss for providing the Lagranto backward trajectory analysis and the meteorological data, and the Deutscher Wetterdienst and Olivier Trollé from MétéoFrance for the radio sounding and ceilometer data. We are grateful to Alastair Williams and his group at ANSTO for the ongoing collaboration as the supplier and supporter of the radon detection system. This research was funded by the Global Atmospheric Watch, Switzerland (MeteoSwiss GAW-CH+ 2018–2021). We acknowledge that the International Foundation High Altitude Research Stations Jungfraujoch and Gornergrat (HFSJG), 3012 Bern, Switzerland, which made it possible for us to carry out our experiment(s) at the High Altitude Research Station at Jungfraujoch, with a special thanks to Claudine Frieden, Prof. Dr. Markus Leuenberger and the custodians Joan and Martin Fischer, Christine and Ruedi Käser, and Daniela Bissig and Erich Furrer. The radon observations at Jungfruajoch and the ceilometer observations at Kleine Scheidegg are supported by the Swiss National Science Foundation (SNSF) as a contribution to the pan-European Integrated Carbon Observation System (ICOS) Research Infrastructure. The continuous aerosol measurements at the Jungfraujoch site are supported by MeteoSwiss in the framework of the Swiss contributions (GAW-CH) to the Global Atmosphere Watch program of the World Meteorological Organization (WMO), and the ACTRIS research infrastructure funded by the Swiss State Secretariat for Education, Research and Innovation (SERI) and by the European Commission under the Horizon 2020 - Research and Innovation Framework Programme, H2020-INFRADEV-2019-2, Grant Agreement number: 871115 (ACTRIS IMP). We thank Prof. Dr. Ulrike Lohmann for her support and enthusiasm. We acknowledge Dr. Heike Wex, Jörg Wieder, Dr. Zane Dedekind, Dr. Larissa Lacher, Dr. Fabian Mahrt, Julie Pasquier, and Dr. Carolin Rösch for useful discussions. For technical support and fabrication, we would like to thank Dr. Michael Rösch and Marco Vecellio, whose expertise greatly helped to improve the instrumentation. This paper was edited by Prof. Lynn M. Russell and reviewed by Dr. Paul DeMott and two anonymous referees.

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
