# Peer review of "The contribution of Saharan dust to the ice nucleating particle concentrations at the High Altitude Station Jungfraujoch (3580 m a.s.l.), Switzerland"

_Atmospheric Chemistry and Physics, 2021_

## Author Comment (AC1)

Reviewer comments are reproduced in **bold** and author responses in normal typeface; extracts from the original manuscript are presented in *red italic*, and from the revised manuscript in *blue italic*.

**This is a very interesting study using the automated HINC instrument, which is packed with useful results and honest (refreshing) caveats, limited only perhaps by the assessment being made at a relatively low temperature (243K, for conditions that emphasize water droplet co-activation and expected immersion freezing). That is a modest critical comment, really, as it does not terribly limit the insights reflected in the deep analyses conducted. The study reveals important links between ice nucleation at lower temperatures and Saharan dust transports in this region, an apparently ubiquitous and strong influence, made further relevant by the observation that higher altitude dust appears to be drawn down via virga following likely heterogeneous ice formation at cirrus levels. The authors emphasize that this raises the potential issue of pre-activation and the known inability of most INP instrumentation for assessing this without special preconditioning of sampled air (needs more discussion). I wonder if this behavior does not also point to an important potential link between dust and precipitation in storms where deep clouds form over the region under dust transports. Precedence for this occurrence already exists in the literature over other regions that are more distant from major mineral dust sources. If so, transported dust INPs at high altitudes could more importantly impact precipitation (via ice sedimentation effects) than one might guess based on their characteristics of activating only at lower temperature, versus some other INP types that are imagined to have a greater impact on mixed-phase clouds due to their higher activation temperatures. I feel that a potential missed opportunity in this paper is relating the INP data more directly to aerosol size distribution measurements. An important relation to aerosol surface area is only inferred in the paper, and indirectly supported via the relation shown with attenuated backscatter. Closure with actual measured aerosol surface area to show relation to INPs could be useful (not required, just a suggestion). Presently it is only stated that some of the surface area (35 % estimate) is missed by the measurement limitations on capturing all sizes. That seems a minor amount to miss, something that I think should be emphasized as a positive aspect of the real-time, automated methods. I have only an assortment of other specific comments added to this, which I do below in order of appearance.**

We would like to thank Dr. Paul DeMott for reviewing the manuscript, for the valuable comments and compliments and address the comments individually below.

**Line 4: I suggest that this statement is not true in general, and so requires more generalization without attribution of a specific temperature. There are many reports of other influences on ice formation dominant at this temperature that are not only mineral dusts. For example, also soil dusts, which made me think about whether (or why not) this site does or does not see influence for more local/regional soil sources.**

We agree with the reviewer, and changed lines 3-4 (revised manuscript) from the abstract:

*Mineral dust* *has been found* *to be* *one of the most abundant INP in the atmosphere at temperatures colder than 258 K.*

**Line 8: I suggest to mention the method for derivation of Ångström exponent here (nephelometer), just as satellite retrieval is mentioned for mass concentrations.**

We agree with the reviewer, and changed lines 8-10 (revised manuscript) as proposed:

*Using the single scattering albedo Ångström exponent* *retrieved from a nephelometer and an aethalometer, satellite retrieved dust mass concentrations, simulated tropospheric residence times, and the attenuated*

*backscatter signal from a ceilometer as proxies, we detected 26 SDEs, which in total contributed to 17 % of the time span analyzed.*

**Line 15: Can you distinguish mineral from soil dust particles (due to attribution scheme for SDEs)?**

No, we cannot distinguish the two types of dust with the sensing methods used in the current work. Thus, we changed lines 15-16 (revised manuscript) from "mineral dust" to "dust", which is also consistent with the statement in the conclusions. Thank you for catching that.

*We estimate 97 % of all INPs active in the immersion mode at 243 K $S_w$ = 1.04 at the JFJ to be  dust particles.*

**Line 52: This is obviously not critical, but just to note that DeMott et al. (2003) was revised in 2009 to indicate a correction to maximum INP concentrations, reducing them to a few hundred per liter.**

We thank the reviewer for the note, and changed lines 51-52 (revised manuscript) as follows:

*In these studies, mineral dust showed the ability to be ice-active in the deposition or immersion mode with concentrations exceeding 500 to 1000 INP std L−1 at 236.6 to 248 K (DeMott et al., 2003; DeMott et al., 2009; Bi et al., 2019).*

**Lines 52-55: I feel it is necessary to say that Zhao et al. was a modeling study, depending both on proper modeling of aerosols and parameterization of INPs.**

We agree with the reviewer, and changed lines 53-57 (revised manuscript) as proposed:

*Zhao et al. (2021) studied the global contributions to INP concentrations at 248 K using the Community Earth System Model version 2 (CESM2) and found dust over the terrestrial midlatitudes to be the dominating INP species by one to two orders of magnitude higher INP number concentrations at 248 K compared to marine organic aerosols. In general, the contribution of different INP species is expected to vary depending on the implementation of the INP parameterization in the model and aerosol representations in the model.*

**Lines 92-93: "There is no appreciable natural source of mineral dust in proximity of the site." Is it true that there are no regional sources through the year of other dusts, such as agriculture?**

We thank the reviewer for the comment. Given the remote location of the site in the Swiss Alps, the closest agricultural fields are 15 km to the north; ~3000 m lower in altitude and not used to grow crops (no plowing). Given the structure of the valley, the total area of the agricultural fields is very small. For all remaining directions, agricultural fields are further away. We added lines 105-106 (revised manuscript) as follows:

*There is no appreciable natural source of mineral dust in proximity of the site. Potential local sources of arable dust are isolated agricultural fields 15 km north of the JFJ.*

**Lines 109-110:**

**"If the plumes appeared not to originate from within the Planetary boundary layer (PBL), the period for which the plume is observed between 3200 - 4000 m a.s.l. is marked as SDE." Is it always so that origination in the boundary layer is so easily determined, for the entire history of a parcel?**

We thank the reviewer for the excellent comment. No, it is not, also not in this case: the ceilometer does not reflect the history of an air parcel, but only the change of a passing aerosol plume above the measurement site over time. To reduce potential misinterpretation, we changed lines 123-124 (revised manuscript) as follows:

*If  an aerosol plume was detected above the planetary boundary layer (PBL), the period for which the plume is observed between 3200 - 4000 m a.s.l. is marked as a SDE.*

**Section 2.2.2: Could you add to say a little more about the Ångström exponent calculation, and why a negative value clearly marks an SDE? I think it would help the general reader.**

We thank the reviewer for the proposition, and added lines 131-135 (revised manuscript) as proposed:

*The single scattering albedo Ångström exponent (αSSA) is an indicator of aerosol optical properties, which change during the presence of SDEs. Collaud Coen et al. (2004) observed that the exponent of the single scattering albedo during SDEs decreases with wavelengths, which counteracts the usual increasing trend. This is a combined effect of the peculiar spectral dependence of Saharan dust complex refractive index and large particle size. αSSA is retrieved from a nephelometer (Airphoton, IN101) and an aethalometer (MAGEE scientific, AE33) according to Collaud Coen et al. (2004).*

**Lines 174-179: This is an important discussion, perhaps for reasons that are not highlighted. Although there is size limitation on transmission of particles to the HINC, the losses of INPs are not so great. That is, if indeed mineral dust INPs scale with surface area, then a 35 % loss is not so significant in consideration that different INP measurement methods easily disagree at times by numeric factors up to 10. Hence, it would be great to see documentation of this statement with a size distribution typical of SDE periods. Not a requirement, but a suggestion. And could you also state if an upstream impactor is used for the HINC or not as part of INP detection?**

We thank the reviewer for the important comment. We agree that the stated missed fraction of 35 % is low compared to other caveats. Thus, we changed lines 197-198 (revised manuscript):

*This is a noteworthy limitation, as studies have reported a majority of INPs at T > 248 K to be supermicron particles (e.g., Mason et al., 2016; Creamean et al., 2018; Gong et al., 2020), and for a constant ice active fraction INP concentrations scale with total particle surface area for a given temperature (e.g., Connolly et al., 2009; Niemand et al., 2012). However, instrument comparisons typically report substantially larger discrepancies between individual instruments than 35 % (e.g., see Hiranuma et al., 2015; DeMott et al., 2018).*

Furthermore, we add as proposed the following Figure to the appendix of the manuscript to line 551 (revised manuscript):

[Figure]

*Figure A2: Mean particle size distributions at the JFJ during a SDE at 10:00-12:00 UTC on July 10, 2020 (blue line) and between January 1, and December 31, 2020 (black dashed line), both measured with an OAS. Only particles with diameters < 2.5 µm are sampled by HINC-Auto due to the sampling line geometry and flow rates used. All particle >2.5 µm (gray shading) between February 7 and December 31, 2020 contribute during all SDE periods to 35 % to the overall particle surface area.*

Furthermore, we changed lines 189-194 (revised manuscript) as follows to refer to the new Figure and comment on the lack of an impactor:

*Due to the low volume flow of HINC-Auto and the sampling line geometry, particle survival measurements using an optical particle counter (MetOne GT-526S) and an OAS showed limitations when sampling particles d > 2.5 µm (see Appendix A for detailed information), despite the fact that no impactor was used upstream of HINC-Auto. According to particle size distribution measurements from the OAS, particles with d > 2.5 µm contributed during the investigated SDE periods on average to 35 % to the overall particle surface area (see Figure A2 for an example of particle size distributions during a SDE).*

**Section 2.3: The nature of the correction for frost background is unclear. Simple subtraction? Counts exceeding the 1 sigma value of the filtered air value? And in general, it would be useful to repeat how INPs are distinguished from other large particles in the HINC, especially during SDEs. I realize that an entire instrument paper was recently published, but some of these things bear repeating in short form.**

We thank the reviewer for the comment on the nature of the correction for frost background. We changed lines 216-230 (revised manuscript) as follows:

*False-positive counts can arise in HINC-Auto. Frost grows on the chamber walls, breaks off and is detected at the outlet as ice. This happens irrespective of whether ambient or particle-free air is sampled with HINC-Auto. Thus, to correct the measured INP concentrations, the number of frost particles is measured*

*separately and subtracted from the uncorrected INP measurements. This is done by sampling particle-free air for a period of 5 min before and after an ambient air measurement. During these periods, the number of false-positive frost particles are counted and subtracted time-proportional from the ambient air measurement in between. The recorded false-positive counts per unit time follow a Poisson distribution. Therefore, in a fraction of cases more or fewer false-positive counts per unit time are recorded during the particle-free measurements than during the ambient air measurement. This results in fluctuations of the measured INP concentrations even if the true atmospheric INP number concentration was to remain constant. The standard deviation of the resulting probability density function corresponds to the stated counting uncertainty of ± 1 σ, which is provided with INP concentrations stated in the present work. This counting uncertainty is also considered to be the 1 σ limit of detection (1σ-LOD) for a single data point. However, in the present work all background-corrected INP concentrations are retained, including positive values below the 1σ-LOD and negative values. This approach ensures that the random noise in background-corrected INP values caused by subtracting the mean frost particle counts does not introduce a systematic bias in mean or median values, which would occur if data below the 1σ-LOD were discarded.*

To address how we distinguish INP from other large particles in HINC-Auto, we added lines 199-202 (revised manuscript):

*INPs are detected if particles with an optical diameter of ≥ 4.0 µm are counted at the chamber exit. Only particles with a diameter < 2.5 µm are sampled by the chamber, and the maximum expected size of a droplet that activated on a 2.5 µm particle is well below 4 µm. Therefore, the method is robust as only ice crystals grow past the set size threshold of 4 µm in the set conditions. The frequency of the INP measurements is every 20 min (15 min sampling plus 5 min background), corresponding to N = 19561 measurements between February 7 and December 31, 2020. See Brunner and Kanji (2021) for more information on the sampling and derivation of the INP concentrations.*

**Figure 4: Would one infer from this figure that the SDE encompassed primarily boundary layer air transport? Again, I was confused because there was such an emphasis placed on indicating FT origin, and I was expecting that FT would encompass most SDE. Also, in this Figure caption and in the main manuscript text, I suggest writing out the meaning of hcSDE and lcSDE.**

We thank the reviewer for the comment. SDEs are often transported in the FT, but then sediment into the PBL. We quantify the fraction of SDE periods at the site for with FT or BLI was present thus proposing that both scenarios are possible for SDEs. To clarify this, we added lines 318-322 (revised manuscript):

*In former studies, SDEs were reported to be occurring in the FT only (Lacher et al., 2018a); however, our results indicate that FT conditions ($P_{FT}$ ≥ 50 %) made up 14.5 % of the total SDE-time, compared to non-SDE periods, where FT conditions prevailed for 40.5 % of the time.  A smaller FT fraction during SDE periods versus non-SDE periods is expected because of the seasonality of SDEs, with few events in winter, when also FT conditions prevail.*

Furthermore, we added the meaning of hcSDE and lcSDE to the Figure caption as proposed:

*Figure 4. An example period classified as high confidence Saharan dust event (hcSDE) between July 7 and July 11, 2020: …*

And accordingly to the caption of Figure 7:

*Figure 7. Dust$_{CAMS}$ mass concentrations with simultaneously measured INP number concentrations (a), and dustCAMS mass concentrations with simultaneously measured PM10 concentrations in (b), each for periods of high confidence Saharan dust events (hcSDEs), low confidence Saharan dust events (lcSDEs) and non-SDEs, which the corresponding correlation parameters and linear the fit (blue line).*

**Please note, I could not find anywhere else in the paper where these were specifically defined. That is, what makes for attributing hcSDE versus lcSDE?**

The definition of high confidence Saharan dust event (hcSDE) and low confidence Saharan dust event (lcSDE) is provided in lines 98-100 of the original manuscript (now lines 113-114).

**Line 250: Particulate matter means mass concentration? Or total counts in a certain size range?**

We mean mass concentration. We thank the reviewer for the comment, and changed lines 292-293 (revised manuscript) as proposed:

*The mass concentration of particulate matter with an aerodynamic diameter below 2.5 μm (PM2.5) and below 10 μm (PM10) is continuously recorded with a white light optical aerosol spectrometer (Fidas 200, Palas GmbH, Germany).*

**Section 2 general comment: It seems like the CAMS, FLEXPART modeling and CALIOP data are the key to assuredly attributing dust to a Saharan source. That is, an aerosol layer in ceilometer data and Angstrom exponent will only indicate larger particles present from somewhere. I am curious if there were ever indications of arable dust sources reaching JFJ. It seems common in other locales, including mountaintop sites that are perhaps not so elevated. Unfortunately, I cannot offer a reference for that because the separate works I am aware of are still in preparation. It is simply an honest question about whether any such influences were inferred during the non-SDE periods or if any lcSDE events could have captured such.**

We thank the reviewer for the very good question. We are not aware of any work at the JFJ quantifying the contribution of soil dust to the overall aerosol or dust population. We added we added lines 369-385 (revised manuscript) as follows, addressing also the comment within the results section from reviewer #3 regarding the influence of a biogenic coating:

*Thus, 74.7±0.2 % of all INPs were detected during SDEs. Whether this is because of the ice nucleation activity of the mineral dust or because of biological surface features, as proposed by other studies (e.g., see O'Sullivan et al., 2016; Augustin-Bauditz et al., 2016), is outside of the scope of this study. Biological surface features on the dust particles as the predominant cause of the ice nucleation activity of the particles would have two main implications. Firstly, the underlying mechanism leading to ice nucleation might differ, as proteins and other macromolecules could induce the ice nucleation as compared to topological mineral surface features, such as cracks and pores. Secondly, it would raise the question of the source of the dust particles containing ice-active biological surface features to be potentially from dried lake beds in desert regions or agricultural regions that are not differentiated in this work. In addition, a contribution of arable dust cannot be ruled out, however, during the 26 SDEs the modelled FLEXPART particles surface residence times indicated in 24 cases that the air mass had robust surface contact in the Saharan Desert, and in the other two cases, they had weak surface contact. The median INP concentration during SDE periods with FT air masses is 17.3 INP std L$^{-1}$, while with BLI it is 23.7 INP std L$^{-1}$. If we assume arable dust to show a substantially larger signal within the well-mixed PBL than in the FT, we could attribute the difference between median INP concentrations in FT SDE periods and BLI SDE periods to be because of arable dust, which is 6.4 INP std L$^{-1}$. There were three SDEs detected with lower concentrations than 6.4 INP std L$^{-1}$, one with a signal in $\alpha_{SSA}$, one without and one where the nephelometer was offline. If during SDE periods the median contribution of arable dust was 6.4 INP std L$^{-1}$, then this should be also similar during non-SDE periods. However, the median INP concentration during non-SDE periods was only 1.1 INP std L$^{-1}$. We do not see any indications why during SDEs the contribution of arable dust should be substantially larger than during non-SDE periods. If at all, we expect arable dust would contribute to BLI INP concentrations during non-SDE periods.*

**Lines 338-339: I appreciate the strong qualifications added to the conclusion on the major role of mineral dust outside of SDEs for the conditions examined, but I wonder if somewhere here or in the conclusions you could speculate on what other types of measurements (or extensions) could be done to support the major role of mineral dust that you hypothesize here for this site (and by proxy perhaps, these altitudes over the region)? I say that because, in boundary layer measurements, my impression is that mineral dust cannot be assumed as the only influence at 243K, though I will not list the references.**

We thank the reviewer for the constructive comment. We assess that separating the INPs from the aerosol population with subsequent chemical composition analysis can be a way to verify the stated hypothesis. We also have an idea of how to use one of the drawbacks of horizontal CFDCs to achieve the separation of INPs and CCN/remaining aerosols without the need of a virtual counter flow impactor. The idea is published in the Ph.D. thesis of Cyril Brunner, chapter 6.2.3. To elaborate, we added lines 401-406 (revised manuscript) as proposed, addressing also the comment within the results section from reviewer #3 regarding the detection of ice-active biological surface features:

*Note, during non-SDE periods, dust contributed 23 / 25.3 % ≈ 91 %  to the overall INP population. To validate the stated contribution in future studies or investigate the presence of biological surface features causing the ice-activity, we propose to separate INPs from the bulk aerosol population to analyze the chemical composition of the INPs as well as study the surface using scanning electron microscopy. However, to our knowledge, such equipment to separate INPs has not been used in an annual study. Detailed suggestions of how this can be achieved in a long-term automated study by modifying HINC-Auto is presented elsewhere (Brunner 2021), and is beyond the scope of the current manuscript.*

**Lines 355-356: Do natural INP concentrations scale with aerosol surface area?**

No, they generally do not unless an air mass is dominated by a specific INP species. Only for if the INP type remains the same. We thank the reviewer for the comment, and changed lines 421-424 (revised manuscript), addressing also the comment about lines 351-363 (lines 421-427 revised manuscript) from reviewer #2:

*This does not come as a surprise, as the INP concentration is a particle number concentration per volume of air that tends to scale with particle surface area ($\propto r^2$) for an identical INP type or air mass dominated by a certain INP species or with the number concentration of viable dust particles, while dustCAMS provides a mass concentration ($\propto r^3$).*

**Line 434: Regarding these results apparently linking cirrus virga to descent of mineral dust plumes to levels where they can impact ice formation at lower levels, the mystery to me is why the virga connect to regions of apparent higher dust concentrations than ever seem to appear aloft.**

We also were wondering the same but have no conclusive answer. Multiple studies state the top of the Saharan dust layer was at 6000 m a.s.l. and transport to higher altitudes seems improbable. Nonetheless, we detect that the plumes in the ceilometer show substantially elevated INP concentrations, and these plumes often connect to virga. The virga could also be spotted by eye as well at the time of the occasion.

**Lines 440-442: You say as much in the abstract, but I expected a few more words here. It is possible I suppose to explore this topic in a laboratory setting, or by exposing instruments to the natural environment. I suspect that unreported efforts have already been attempted. It will be immensely more difficult to do this in an aircraft setting, where the actual nature of the inlet outside the aircraft would have to be modified in order to not heat particles due to pressure. In any case, I think you should expand the discussion a little bit, for the sake of those who may not have deeply considered how this can be done (or how hard it may be).**

We agree with the reviewer, and changed line 519-526 (revised manuscript) as follows:

*If the pre-activated particles are again exposed to ice supersaturated conditions, spontaneous ice crystal growth can occur at much lower Si than with other INPs (David et al., 2019). To capture and study this effect, design adaptations need to be implemented in our sampling equipment in upcoming studies, as the pore ice sublimates or melts in the heated and dried sampling lines upstream of the INP counter used here. We attempted such a measurement of the number concentration of pre-activated particles in this way, using a supercooled diffusion dryer at the JFJ before directly sampling the ambient particles with HINC, in addition to sampling with a reference chamber with the standard heated inlet. Nevertheless, we were unable to reach any reliable conclusions. In part, due to the clogging of the inlet from frost build-up, the conditions along the trajectory of the potentially pre-activated particles had to be precisely adjusted, such that neither particle sedimentation due to activation, nor pore ice sublimation occurs. Furthermore, the set conditions do not affect all pore sizes similarly, and pore-ice can melt if the pore ice shrinks below the critical size of an ice embryo within the pore with a constrained width. In summary, there were too many parameters that needed to be simultaneously and meticulously controlled.*

**Lines 25: It is unusual to start a sentence with a number. Perhaps write it out.**

Thanks for catching that. We changed line 25 (revised manuscript) as proposed:

*Sixty-three $\pm$ 7 % of global precipitation is initiated via the ice phase (Heymsfield et al., 2020), predominately over land and in the midlatitudes (Mülmenstädt et al., 2015).*

**Line 31: cover clouds → do clouds cover**

Thanks for catching that. We changed line 31 (revised manuscript) as proposed:

*Not only do clouds cover 68 % of Earth's surface…*

**Lines 69: Suggest to spell out 20 "minutes" for the INP measurement time resolution.**

Thank you for the proposition. We changed line 80 (revised manuscript) accordingly:

*During this time, continuous high-resolution (20 minutes) online INP measurements were performed for the first time at the JFJ.*

**Lines 90: hosts → has hosted**

Thanks for catching that. We changed line 103 (revised manuscript) as proposed:

*The JFJ has hosted long-term aerosol measurements for more than 30 years …*

**Lines 119: previously is misspelled.**

Thanks for catching that. We changed line 136 (revised manuscript) as proposed:

*This is longer than the previously used 4 hours in Collaud Coen et al. (2004) in order to decrease the number of false or suspicious signals due to construction work at the JFJ (see below).*

**Lines 127: with is misspelled.**

Thanks for catching that. We changed line 144 (revised manuscript) as proposed:

*Gueymard and Yang (2020) performed a worldwide validation of the aerosol optical depth and Ånsgtröm exponent from CAMS and MERRA-2 with ground-based AERONET stations over the period 2003–2017.*

**Lines 272: where → were**

Thanks for catching that. We changed line 316 (revised manuscript) as proposed:

*…, and the highest concentrations measured were half as high as those measured in the Saharan Air Layer in Tenerife…*

**Lines 425: Probably do not need the word "for".**

Thanks for catching that. We changed line 502 (revised manuscript) as proposed:

*The analysis showed  trajectories to originate from the Saharan desert within Algeria,…*

**Lines 461: Suggest rewrite, e.g., "and sediment to lower altitudes where they sublimate in drier air…"**

Thanks for catching that. We changed line 545 (revised manuscript) as proposed, addressing also the comment of reviewer #2 about line 415 (line 485 in the revised manuscript):

*We found examples of SDEs with upstream virga from altitudes above 8000 m a.s.l., which led to the hypothesis of INPs being transported  to the midlatitudes, where they nucleate ice at altitudes above 5500 m a.s.l. and sediment to lower altitudes where they sublimate in drier air , and act as INPs at these lower altitudes.*

**Lines 463: loose → lose**

Thanks for catching that. We changed line 547 (revised manuscript) as proposed:

*This hypothesis will be subject of a future study, as pre-activated INPs lose their pore ice in the heated and dried sampling lines used in this study.*

References

Brunner, C.: Monitoring of Ice Nucleating Particles (INP) at the Jungfraujoch: Automation of the Horizontal Ice Nucleation Chamber for Continuous INP Monitoring (Doctoral dissertation), chapter 6.2.3, available from ETH Zurich research collection, doi: 10.3929/ethz-b-000493877, 2021.

DeMott, P. J., Sassen, K., Poellot, M. R., Baumgardner, D., Rogers, D. C., Brooks, S. D., Prenni, A. J., and Kreidenweis, S. M.: Correction to "African dust aerosols as atmospheric ice nuclei", Geophys. Res. Lett., 36, L07808, doi:10.1029/2009GL037639, 2009.

DeMott, P. J., Möhler, O., Cziczo, D. J., Hiranuma, N., Petters, M. D., Petters, S. S., Belosi, F., Bingemer, H. G., Brooks, S. D., Budke, C., Burkert-Kohn, M., Collier, K. N., Danielczok, A., Eppers, O., Felgitsch, L., Garimella, S., Grothe, H., Herenz, P., Hill, T. C. J., Höhler, K., Kanji, Z. A., Kiselev, A., Koop, T., Kristensen, T. B., Krüger, K., Kulkarni, G., Levin, E. J. T., Murray, B. J., Nicosia, A., O'Sullivan, D., Peckhaus, A., Polen, M. J., Price, H. C., Reicher, N., Rothenberg, D. A., Rudich, Y., Santachiara, G., Schiebel, T., Schrod, J., Seifried, T. M., Stratmann, F., Sullivan, R. C., Suski, K. J., Szakáll, M., Taylor, H. P., Ullrich, R., Vergara-Temprado, J., Wagner, R., Whale, T. F., Weber, D., Welti, A., Wilson, T. W., Wolf, M. J., and Zenker, J.: The Fifth International Workshop on Ice Nucleation phase 2 (FIN-02): laboratory intercomparison of ice nucleation measurements, Atmos. Meas. Tech., 11, 6231–6257, https://doi.org/10.5194/amt-11-6231-2018, 2018.

Hiranuma, N., Augustin-Bauditz, S., Bingemer, H., Budke, C., Curtius, J., Danielczok, A., Diehl, K., Dreischmeier, K., Ebert, M., Frank, F., Hoffmann, N., Kandler, K., Kiselev, A., Koop, T., Leisner, T., Möhler, O., Nillius, B., Peckhaus, A., Rose, D., Weinbruch, S., Wex, H., Boose, Y., DeMott, P. J., Hader, J. D., Hill, T. C. J.,

Kanji, Z. A., Kulkarni, G., Levin, E. J. T., McCluskey, C. S., Murakami, M., Murray, B. J., Niedermeier, D., Petters, M. D., O'Sullivan, D., Saito, A., Schill, G. P., Tajiri, T., Tolbert, M. A., Welti, A., Whale, T. F., Wright, T. P., and Yamashita, K.: A comprehensive laboratory study on the immersion freezing behavior of illite NX particles: a comparison of 17 ice nucleation measurement techniques, Atmos. Chem. Phys., 15, 2489–2518, https://doi.org/10.5194/acp-15-2489-2015, 2015.

Lacher, L., DeMott, P. J., Levin, E. J. T., Suski, K. J., Boose, Y., Zipori, A., Herrmann, E., Bukowiecki, N., Steinbacher, M., Gute, E., Abbatt, J. P., Lohmann, U., and Kanji, Z. A.: Background free-tropospheric ice nucleating particle concentrations at mixed-phase cloud conditions, Journal of Geophysical Research: Atmospheres, 123, 10,506–10,525, https://doi.org/10.1029/2018JD028338, 2018a.

---

## Author Comment (AC2)

Reviewer comments are reproduced in **bold** and author responses in normal typeface; extracts from the original manuscript are presented in *red italic*, and from the revised manuscript in *blue italic*.

The authors conduct measurements of INP from February 2020 to December 2020 at JFJ. The INP measurements are constrained to a temperature of 243K and a saturation ratio of 1.04. They classify the INP according to whether or not Saharan dust events were present. The classification is based on four criteria: single scattering albedo; satellite retrievals of dust mass concentration; modelled tropospheric residence times; the backscatter signal from a ceilometer. 14 dust events of high confidence (hcSDEs) were classified where each of the four criteria for Saharan dust were met and 12 events of lower confidence (lcSDEs) were classified where at least one of the four criteria were met. The authors show that INP concentrations increase by generally one to two orders of magnitude during the periods of dust events. They also find some evidence for dust influence in the absence of an identified event. I find the main aspects of the work to be sound and useful. I think the interpretation of the results, initially sound, is carried a bit farther than warranted, reflected in my comments 12-17 below. Overall, I feel the paper could be suitable for publication subject to some revisions.

We would like to thank the reviewer for their valuable comments and address the comments individually below.

Lines 3-4: These two sentences might be better reversed in order, as I find the second seems to contradict the first.

We agree with the reviewer, and changed lines 3-5 (revised manuscript) from the abstract as follows, addressing also the comment about line 4 (line 4 revised manuscript) from reviewer #1:

However, the extent of the abundance and distribution of INPs remains largely unknown. Mineral dust has been found to be one of the most abundant INP in the atmosphere at temperatures colder than 258 K. However, the extent of the abundance and distribution of INPs remains largely unknown.

**Lines 33-35: Again, slightly contradictory statements: Mixed phase exist between 273K and 235K, yet most clouds warmer than 253-258K are ice free.**

We thank the reviewer for the comment. We changed line 33 (revised manuscript) as follows:

Mixed-phase clouds theoretically can exist between 273 K and ~ 235 K. Depending on the measurement location, in-situ measurements revealed that only approximately half of the clouds contain the liquid phase when at 253 to 258 K, while the warmer clouds are mostly ice free (e.g., Korolev et al., 2003; Verheggen et al., 2007; Kanitz et al., 2011).

**Lines 41-59: I suggest mentioning the importance of mineral dust as INP before discussing the sources and transport of dust.**

We acknowledge the reviewer's comments, however we feel it is better to first address the global distribution and ubiquity of mineral dust in the atmosphere before narrowing down to the INP impacts of mineral dust. As such, we leave the text in lines 41-61 (revised manuscript) the same as in the original manuscript.

**Line 73: Remove "Besides".**

We agree with the reviewer, and changed line 85 (revised manuscript) as proposed:

This allows to analyze whether all SDEs show an increased INP number concentration, as previous studies imply (Chou et al., 2011; Boose et al., 2016b; Lacher et al., 2018a). Besides, oOur data indicate that signals from Light Detection and Ranging (LIDAR) ceilometers can be used to infer INP concentrations, as reported in other studies using depolarization channel LIDARs (Mamouri and Ansmann, 2015; Ansmann et al., 2019).

**Line 250: What metric of particulate matter is recorded?**

We were referring to mass concentration. We thank the reviewer for the comment, and changed line 292 (revised manuscript) as follows:

The mass concentration of particulate matter with an aerodynamic diameter below 2.5  $\mu$ m (PM2.5) and below 10  $\mu$ m (PM10) is continuously recorded with a white light optical aerosol spectrometer (Fidas 200, Palas GmbH, Germany).

**Line 256: Please clarify what you mean by "and involved microphysics of dust".**

We thank the reviewer for the comment, and changed lines 298-300 (revised manuscript) as follows:

To assess the atmospheric transport of dust and the presence and retrieved phase of nearby clouds in the case study (see subsection 3.3), post-processed lidar data from the Cloud-Aerosol Lidar with Orthogonal Polarization (CALIOP) instruments of the Cloud-Aerosol Lidar and Infrared Pathfinder Satellite Observations (CALIPSO) satellites was used.

**Lines 264-270: Perhaps clarify that the median of all single events is the median of the individual event medians, and that the collective refers to the median of all INP concentrations during the SDEs.**

We thank the reviewer for the comment, and changed lines 307-313 (revised manuscript) as proposed:

The median INP concentrations of the individual event medians was  $15.3 \pm 1.2$  INP std L-1. Analogously, the medians of all single event 25 %, 50 %, and 75 % quartile INP concentrations were  $9.1 \pm 1.1$ ,  $15.3 \pm 1.2$ , and  $32.4 \pm 1.4$  INP std L-1 (event-based; see Table 1, and Table A1 for more detail), generally one order of magnitude higher than the median INP concentration of  $1.1 \pm 1.0$  INP std L-1 during periods without SDEs (non-SDE). All SDE quartiles except the 25 % quartile exceed the 95th percentile during non-SDE conditions. Considering the collective INP concentration during all SDE periods, the median of all INP concentrations during all SDEs combined increases to  $22.5 \pm 1.4$  INP std L-1, rendering the longer dust events generally to contain more ice-active particles. This median concentration is consistent with previously reported values at the JFJ of 26.1 INP std L-1 (Lacher et al., 2018a).

**Lines 277-279: They clearly differ based on the stated uncertainties, yet you say the difference is not significant. Please explain.**

We thank the reviewer for the comment, and changed lines 322-325 (revised manuscript) as follows:

The observed INP concentrations during SDEs were lower in FT conditions compared to periods with BLI with median concentrations of  $17.3 \pm 1.1$  and  $23.7 \pm 1.5$  INP std L-1, respectively, however, they did not significantly differ, as the median INP concentrations of one class does not exceed the interquartile range INP concentrations of the other class and vice-versa.

Lines 297-298: With only 12 lcSDE cases, the statistics for these events cannot be strong, unless one criterion was dominant.

The statement on line 297-298 (original manuscript, now lines 343-344) is based on all 26 SDEs detected. I.e., we believe that increased INP concentrations can be detected as long as one of the tracers exhibits and SDE. While we are limited with how many SDE occurred during the year, and more statistics are of course always desirable, this is a substantial leap forward in the statistics of the number of events detected compared to previous studies where single field campaigns on the order of 2-6 weeks were used to inform the impact of SDEs on INP concentrations.

**Lines 310-311: Maybe, you can't say this is true, or even implied, without some sort of chemical ID.**

We agree, and changed lines 356-357 (revised manuscript) as proposed:

This could mean that the kind of INPs detected during SDE also contributes to the overall INP population during non-SDE periods, but chemical analysis would be necessary to categorically conclude this.

Figure 6b – You mention having normalized the area under each curve. I suggest adding that this does not allow the "SDE" plus "no SDE" curves to equate to "all".

We agree with the reviewer, and changed Figure 6 caption as proposed:

Figure 6. Frequency distributions of INP concentrations (a); and dustCAMS (b) between February 7 and December 31, 2020 (solid black), for all classified SDEs (lcSDE and hcSDE, green) and for periods without SDEs (pink). A log-normal curve with stated curve parameter in (a) has been fitted to the frequency distribution of all INP concentrations (dashed blue). The area under each frequency distribution is normalized to unity, which does not allow the sum of the areas below the SDE and non-SDE frequency distributions to be equal to the area below the frequency distribution of all classified SDEs.

Lines 334-339: This a simple back-of-the-envelope calculation. The inclusion of uncertainties of 0.1% and 0.2% suggests otherwise and I think is inappropriate. I suggest reducing this discussion to something like "With our assumptions, we estimate that about 23% of the INPs measured during non-SDE periods were dust-related.".

We agree with the reviewer, and changed lines 396-401 (revised manuscript) as proposed:

If 76.5% of the dust is responsible for 74.2±0.2% of the INPs (74.7±0.2% - 0.5%), and 23.5% of all dustCAMS was advected to the JFJ during non-SDE periods, we estimate that about 23% of the INPs measured during non-SDE periods were dust-related with our assumption that this dust would proportionally contribute 74.2/ 76.5 × 23.5% = 22.8±0.1% of INPs during non SDE measured at the JFJ, assuming a constant mass fraction of dust acts as INPs. Therefore, the total contribution of dust to the INP population measured at the JFJ at T = 243 K and saturation ratio of Sw = 1.04 is estimated to be 74.2±0.2% + 23 22.8±0.1% ≈ 97.0±0.3%. Note, during non-SDE periods, dust contributed 23 / 25.3% ≈ 91% 22.8 / 25.3% = 90.1% to the overall INP population.

Lines 351-363 - The potential for correlation of INP with dustcams is based on Figure 6 showing consistency between the INP and dustcams distribution. However, the dustcams distribution for the SDE cases does not exhibit a log normal, which suggests that the sizes of dust particles vary, perhaps substantially. One consequence of that is the number concentrations of dust particles will not necessarily scale with dust mass. The authors note that INP scales with r^2, but that is largely related to the process of ice nucleation. Actual INP concentrations likely also scale with simply r or just the number of viable dust particles. Significant improvement of this discussion is needed.

We now clarify the discussion to better reflect the reviewer concerns as follows (lines 421-427 revised manuscript), addressing also the comment about lines 355-356 (lines 421-424 revised manuscript) from reviewer #1:

This does not come as a surprise, as the INP concentration is a particle number concentration per volume of air that tends to scale with particle surface area ( $\propto r^2$ ) for an identical INP type or air mass dominated by a certain INP species, or with the number concentration of viable dust particles, while dustCAMS provides a

mass concentration ( $\propto r^3$ ). Given the distribution of the dust particles during the SDEs is not log-normal (see Figure 6b) suggests that the size of dust particles varies, and thus, the number concentrations will not scale with dust mass. Therefore, dustCAMS is also compared to PM10, which both are in units of mass per volume of air.

**Line 378 and Figure 9 – The "R^2" in Figure 9c looks to be 0.826. Is that correct?**

We thank the reviewer for the question. Indeed, R2 in Figure 9c is 0.826. We increased the font size of the in-figure text to improve readability. However, we note that what is referred to in line 378 (now line 448) is not the R2 but the Spearman's rank correlation coefficients.

**Lines 393-394: Please explain the evidence for a connection between the dust and the virga, which to me looks tenuous at best.**

As stated in Line 414, we only formulate the hypothesis, that the dust are residual particles after the sublimation of the virga, and state in lines 431-434 that the hypothesis remains to be proven in future work. Thus, we do not present any evidence, but rather want to share our observations, that in two case studies Saharan dust plumes, which showed increased INP concentrations compared to background concentrations, are collocated with the low altitude part of virga with higher attenuated backscatter signals than the Saharan dust plumes, as illustrated in Figures 11 and 12. Here extracts from the Figures, where the green dashed lines indicate the collocation of the boundaries of the virga and the Saharan dust plumes:

---

## Author Comment (AC3)

Reviewer comments are reproduced in **bold** and author responses in normal typeface; extracts from the original manuscript are presented in *red italic*, and from the revised manuscript in *blue italic*.

This study provides interesting insights into the linkage between mineral dusts from the Sahara and ice nucleation activity at low temperature at the Jungfraujoch station. By using a wide array of instruments, techniques, and model data the authors were able to assign Sahara dust events and link it to INP concentrations. Even though many assumptions were necessary for the data curation, the authors clearly stated the limitations of each technique. The authors further hypothesize possible dust transport pathways within the atmosphere. Generally, the article is well written, concise and the results are of general interest for the scientific community. The provided data will improve the knowledge of the field and are important for further model studies on clouds and climate. However, I suggest minor revisions before publication:

We would like to thank the reviewer for their comments and compliments on the manuscript, and address the comments individually below.

Can you explain in the introduction in more detail why you have chosen 243 K and Sw of 1.04 as the parameters for your measurements? I assume it is due to general mixed phase conditions. However, why is this particular temperature area so important for ice nucleation including mineral dust and not e.g. 250K? Please elucidate in more detail so a general reader can better follow the content.

We thank the reviewer for the comment, and changed lines 70-80 (revised manuscript) as follows:

In this work, we investigate and quantify the INP concentrations at 243 K and  $S_w = 1.04$  (immersion freezing). These conditions were chosen to align with previous INP measurements at the JFJ between 2014 and 2017 (Boose et al. 2016a, Lacher et al., 2018a). Ice formation in stratiform mixed-phase clouds is frequently observed close to the cloud top where temperatures of 243 K are a common lower bound cloud top temperature of mixed-phase clouds in central Europe (e.g., Bühl et al., 2016). In addition, 243 K is the warmest temperature where the instrument's signal-to-noise ratio allows for statistically acceptable data analysis when the sampling site is located in a remote region such as mountain top stations or the Arctic without using an aerosol concentrator. There is an uncertainty of relative humidity and variation in the vertical position of the particles within the aerosol layer in the chamber (DeMott et al., 2015; Brunner and Kanji, 2021) amounting to  $S_w + 0.007$  and - 0.009 and  $\pm 1.11$  K at 243 K and set  $S_w$  of 1.04. To ensure that the entire sample layer experiences  $S_w > 1.0$ , a nominal  $S_w = 1.04$  was chosen. All INP concentrations were measured at the JFJ during all SDEs between February 7 and December 31, 2020.

**Line 116: I think it would improve the manuscript and makes it clearer to the reader if you could explain the Single scattering albedo Angström exponent in more detail.**

We thank the reviewer for the proposition, and added lines 131-135 (revised manuscript) as follows:

The single scattering albedo Ångström exponent ( $\alpha$ SSA) is an indicator of aerosol optical properties, which change during the presence of SDEs. Collaud Coen et al. (2004) observed that the exponent of the single scattering albedo during SDEs decreases with wavelengths, which counteracts the usual increasing trend. This is a combined effect of the peculiar spectral dependence of Saharan dust complex refractive index and large particle size.  $\alpha$ SSA is retrieved from a nephelometer (Airphoton, IN101) and an aethalometer (MAGEE scientific, AE33) according to Collaud Coen et al. (2004).

**Line 203: The classification of air masses comes a little bit abrupt. Could you mention that at the of the introduction where you summarize the investigations? (just a suggestion)**

We thank the reviewer for the proposition and added lines 81-85 (revised manuscript):

In this work, we investigate and quantify the INP concentrations at 243 K and Sw = 1.04 (immersion freezing) at the JFJ during all SDEs between February 7 and December 31, 2020. During this time, continuous high-resolution (20 minutes) online INP measurements were performed for the first time at the JFJ. Because the data are not tied to single field campaigns in active SDE seasons, it also includes SDEs measurements in seasons where SDEs are infrequent. This allows to analyze whether all SDEs show an increased INP number concentration, as previous studies imply (Chou et al., 2011; Boose et al., 2016b; Lacher et al., 2018a). Furthermore, the classification of SDEs is based on four distinct tracers (see Section 2.2) and analyzed with regard to the type of air mass present at the site, i.e., free tropospheric air or boundary layer intrusions (see Section 2.4). Our data indicate that signals from Light Detection and Ranging (LIDAR) ceilometers can be used to infer INP concentrations, as reported in other studies using depolarization channel LIDARs (Mamouri and Ansmann, 2015; Ansmann et al., 2019). In contrast to Mamouri and Ansmann (2015) and Ansmann et al. (2019), the topographic setup of the present study allowed for the ceilometer to scan the same altitude that the INP concentrations were measured at. Estimating the INP concentrations from the ceilometer backscatter signals from all ceilometer stations across Europe allows us to (back-)track the aerosol masses with enhanced INP concentrations and look into their atmospheric pathway, which we demonstrate in a case study. Finally, the contribution of (Saharan) dust to the INP concentration is estimated.

Figure 4: The figure gives a good overview of the collected data to assign a dust event and compare it to INP concentrations. The caption reveals much information. However, I miss an explanation of the figure in the continuous text.

We thank the reviewer for the comment. We added the following explanations to lines 274-290 (revised manuscript):

Figure 4 shows an example of a SDE with all introduced tracers, the air mass type and the INP concentration. example of a SDE with all at KSE showed an increase from background levels to  $\sim 1 \text{ m}^{-1} \text{sr}^{-1}$  at altitudes similar to the JFJ after midnight on July 10, indicating the presence of an aerosol plume. At 11:00 UTC, the signal of the aerosol plume was attenuated by the low-level clouds during the remaining period of the plume event.  $\alpha_{SSA}$  decreased to below zero with decreasing wavelength after 2:00 UTC on July 10 (Figure 4c). The signal becomes less separated with  $\alpha_{SSA}$  above zero at 15:00 UTC and noisy after midnight of July 11, indicating the end of the SDE according to this tracer. DustCAMS mass concentrations exceeded the threshold concentrations on July 9 at 23:00 UTC, as shown in Figure 4d, peaking at 3:00 UTC of July 10 with 19.6  $\mu$ g m-3, followed by a decay, until falling below the threshold at 20:00 UTC. FLEXPART particle surface residence times in Figure 4e indicate between 3:00 UTC on July 10 and 5:00 UTC on July 11 that the air mass is expected to have had ground contact over the Saharan domain. Following all four tracers showing a signal, the SDE was classified as hcSDE. The INP concentrations show an increase from background INP concentrations on July 9 at 23:00 UTC (Figure 4a), to concentrations above 200 INP std L-1, followed by lower concentrations coinciding with  $\alpha_{SSA}$  exceeding zero for one hour. After a brief, second increase in INP concentrations, a decline to background levels at midnight of July 7 followed. No tracer shows an identical onset and decline as observed in the INP concentrations, however,  $\alpha_{SSA}$  was the closest. The BLI air masses were present during the SDE, with 222Rn initially at FT levels at 4:00 UTC on July 10, followed by a rapid change to BLI levels. At the end of the SDE,  $N_{90}$  indicated FT conditions, while 222Rn still pointed to BLI, resulting in the air mass being classified as BLI. During this SDE, all tracers showed clear signals, however, with different start and end times. Note the increase in INP concentrations during the SDE.

The results point out that further characterizations are necessary in order to estimate the role of mineral dust and other INPs on microphysical processes in the atmosphere. You mentioned in the manuscript in one sentence (line 321) that you cannot draw any conclusions on the influence of biogenic coating.

**However, I wonder whether this would be an important information? I think the manuscript would benefit if this would be discussed and commented in more detail.**

We thank the reviewer for the valuable comment. We agree with the comment and changed lines 369-382 (revised manuscript) as follows, addressing also the comment about section 2 from reviewer #1 regarding the influence of arable dust:

Thus, 74.7±0.2% of all INPs were detected during SDEs. Whether this is because of the ice nucleation activity of the mineral dust or because of biological surface features, as proposed by other studies (e.g., see O'Sullivan et al., 2016; Augustin-Bauditz et al., 2016), is outside of the scope of this study. Biological surface features on the dust particles as the predominant cause of the ice nucleation activity of the particles would have two main implications. Firstly, the underlying mechanism leading to ice nucleation might differ, as proteins and other macromolecules could induce the ice nucleation as compared to topological mineral surface features, such as cracks and pores. Secondly, it would raise the question of the source of the dust particles containing ice-active biological surface features to be potentially from dried lake beds in desert regions or agricultural regions that are not differentiated in this work. In addition, a contribution of arable dust cannot be ruled out, however, during the 26 SDEs the modelled FLEXPART particles surface residence times indicated in 24 cases that the air mass had robust surface contact in the Saharan Desert, and in the other two cases, they had weak surface contact. The median INP concentration during SDE periods with FT air masses is 17.3 INP std  $L^{-1}$ , while with BLI it is 23.7 INP std  $L^{-1}$ . If we assume a able dust to show a substantially larger signal within the well-mixed PBL than in the FT, we could attribute the difference between median INP concentrations in FT SDE periods and BLI SDE periods to be because of arable dust, which is 6.4 INP std L-1. There were three SDEs detected with lower concentrations than 6.4 INP std L-1, one with a signal in  $\alpha_{SSA}$ , one without and one where the nephelometer was offline. If during SDE periods the median contribution of arable dust was 6.4 INP std L-1, then this should be also similar during non-SDE periods. However, the median INP concentration during non-SDE periods was only 1.1 INP std L-1. We do not see any indications why during SDEs the contribution of arable dust should be substantially larger than during non-SDE periods. If at all, we expect arable dust would contribute to BLI INP concentrations during non-SDE periods.

**In addition, how could you improve future studies to target this issue?**

We thank the reviewer for the comment. We added lines 401-406 (revised manuscript), which also address the comment of reviewer #1 concerning lines 338-339:

Note, during non-SDE periods, dust contributed 23 / 25.3 %  $\approx$  91 %  $\frac{22.8}{25.3\%} = 90.1\%$  to the overall INP population. To validate the stated contribution in future studies or investigate the presence of biological surface features causing the ice-activity, we propose to separate INPs from the bulk aerosol population to analyze the chemical composition of the INPs as well as study the surface using scanning electron microscopy. However, to our knowledge, such equipment to separate INPs has not been used in a continuous annual study. Detailed suggestions of how this can be achieved in a long-term automated study by modifying HINC-Auto is presented elsewhere (Brunner 2021), and is beyond the scope of the current manuscript.

**Line 82: Jungfraujoch (JFJ) – abbreviation was already introduced above.**

We thank the reviewer for catching that, and changed line 95 (revised manuscript) as follows:

The Sphinx observatory at the JFJ Jungfraujoch (JFJ) is located on a saddle between Mt. Mönch and Mt. Jungfrau in the Swiss alps (46.330° N, 7.590° E) at an altitude of 3580 m a.s.l. (see Fig. 1).

Line 88: space is missing between the number and the unit (check throughout the whole paper).

Thank you for catching that. We have now included a space between the number and percent symbol throughout the manuscript.

**Line 89: m/s or m s-1 – only use one style**

We thank the reviewer for catching this, and changed line 101 (revised manuscript) as follows:

The principal local wind directions between February and December 2020 were 320° (NW) for 62 %, and 150° (SE) for 27 % of the time, while calm wind situations below  $1 \text{ m s}^{-1}$  had a frequency of 11 %.

**Line 119: [...] more 'than' 6 consecutive [...]**

We thank the reviewer for catching this, and changed line 136 (revised manuscript) as follows:

A SDE is detected if the  $\alpha_{SSA}$  is negative for more than 6 consecutive hours.

**Line 119: This is longer than the 'previously' [...] (?)**

Thanks for catching that. We changed line 136 (revised manuscript) as proposed:

This is longer than the previously used 4 hours in Collaud Coen et al. (2004) in order to decrease the number of false or suspicious signals due to construction work at the JFJ (see below).

**Line 127: [...] MERRA-2 'with' [...]**

Thanks for catching that. We changed line 144 (revised manuscript) as proposed:

*Gueymard and Yang (2020) performed a worldwide validation of the aerosol optical depth and Ånsgtröm exponent from CAMS and MERRA-2 with ground-based AERONET stations over the period 2003–2017.*

**Line 135/136: meters were abbreviated before**

We thank the reviewer for the comment, and changed lines 152-155(revised manuscript) as follows:

For the classification of SDEs we used the dust reanalysis data from CAMS (dustCAMS) at 1000 m above the surface in hourly resolution and units of  $\mu g$  m-3, accessed via the Copernicus Atmosphere Data Store. 1000 m above surface was chosen to be closest to the real altitude of the JFJ and accounting for the smoothed surface elevations in the model domain due to the coarse grid spacing.

**Line 227: introduce the abbreviations and delete them below (line 231/232); Carbon dioxide (CO) and reactive nitrogen (NOy)**

We thank the reviewer for the comment, and changed lines 254-260 (revised manuscript) as proposed:

A further tracer of BLI is the ratio of total reactive nitrogen ( $NO_y$ ) to carbon monoxide (CO) according to Zanis et al. (2007). Both  $NO_y$  and CO are emitted from anthropogenic sources, however,  $NO_y$  reacts and decays on the order of days, while CO can be considered inert within the same time period. Consequently, the ratio of  $NO_y$  to CO decreases with increasing ageing time, leading to smaller ratios found in the FT compared to BLI (Zanis et al., 2007). Carbon monoxide (CO) CO is continuously measured with a Cavity Ringdown Spectrometer at the JFJ (Zellweger et al., 2019). Total reactive nitrogen (NOy)  $NO_y$  was measured until March 2020 with a chemiluminescence detector after conversion to NO on a heated gold catalyst (573 K) in the presence of CO as a reducing agent (Pandey Deolal et al., 2012).

**Figure 4 caption: superscript '222'-Radon**

Thanks for catching that. We changed Figure 4 caption as proposed:

(f) The probability for free tropospheric (FT) or boundary layer intruded (BLI) air mass to be sampled at the JFJ for the left hand side factor of equation (1), dependent on the 222Radon concentrations (222Rn, blue line) and the right hand side factor, dependent on the particle concentrations with a mobility diameter larger than 90 nm (N90, red line) as well as their product, the probability of the sampled air to be of free tropospheric origin (PFT, black dashed line).

**Line 253: '85°' is missing a degree sign**

Thanks for catching that. We changed line 295 (revised manuscript) as proposed:

The spectrometer measures the intensity of single particles at an angle between 85° and 95° and infers the particle diameter solving the inverse Mie problem.

**Table 1: All INP 'concentrations' missing the letter s in the end**

Thanks for catching that. We changed Table 1 caption as proposed:

All INP concentrations are in units of INP std L-1.

**Line 302: Figure or Fig.? chose one style**

We thank the reviewer for the comment, and changed line 350 (revised manuscript) as follows:

[...] (blue dashed line in Figure 6a; [...]

**Line 349: I guess at the end of the sentence it should be 'can'?**

Indeed. We thank the reviewer for catching that and changed lines 416 (revised manuscript) as proposed:

Also, at measurement locations further away from the dust source than the JFJ is, or locations closer to local sources in the PBL, the contribution of dust on the total INP population can be expected to be much smaller.

**References**

Boose, Y., Kanji, Z. A., Kohn, M., Sierau, B., Zipori, A., Crawford, I., Lloyd, G., Bukowiecki, N., Herrmann, E., Kupiszewski, P., Steinbacher, M., and Lohmann, U.: Ice Nucleating Particle Measurements at 241 K during Winter Months at 3580 m MSL in the Swiss Alps, Journal of the Atmospheric Sciences, 73, 2203–2228, https://doi.org/10.1175/JAS-D-15-0236.1, 2016a.

Bühl, J., Seifert, P., Myagkov, A., & Ansmann, A.: Measuring ice- and liquid-water properties in mixed-phase cloud layers at the Leipzig Cloudnet station. Atmospheric Chemistry and Physics, 16(16), 10,609–10,620. https://doi.org/10.5194/acp-16-10609-2016, 2016. Brunner, C.: Monitoring of Ice Nucleating Particles (INP) at the Jungfraujoch: Automation of the Horizontal Ice Nucleation Chamber for Continuous INP Monitoring (Doctoral dissertation), chapter 6.2.3, available from ETH Zurich research collection, doi: 10.3929/ethz-b-000493877, 2021.

Brunner, C. and Kanji, Z. A.: Continuous online monitoring of ice-nucleating particles: development of the automated Horizontal Ice Nucleation Chamber (HINC-Auto), Atmospheric Measurement Techniques, 14, 269–293, https://doi.org/10.5194/amt-14-269-2021, 2021.

DeMott, P. J., Prenni, A. J., McMeeking, G. R., Sullivan, R. C., Petters, M. D., Tobo, Y., et al.: Integrating laboratory and field data to quantify the immersion freezing ice nucleation activity of mineral dust particles. Atmospheric Chemistry and Physics, 15(1), 393–409. https://doi.org/10.5194/acp-15-393-2015, 2015.

Lacher, L., DeMott, P. J., Levin, E. J. T., Suski, K. J., Boose, Y., Zipori, A., Herrmann, E., Bukowiecki, N., Steinbacher, M., Gute, E., Abbatt, J. P., Lohmann, U., and Kanji, Z. A.: Background free-tropospheric ice nucleating particle concentrations at mixed-phase cloud conditions, Journal of Geophysical Research: Atmospheres, 123, 10,506–10,525, https://doi.org/10.1029/2018JD028338, 2018a.

---

## Author Response (AR2)

Reviewer comments are reproduced in **bold** and author responses in normal typeface; extracts from the original manuscript are presented in *red italic*, and from the revised manuscript in *blue italic*.

**As I was already convinced that the paper was nearly suitable for publication, I have only a few comments in response to the revisions.**

We would like to thank Dr. Paul DeMott for reviewing the revised manuscript, for the valuable comments and address the comments individually below.

**1. Figure A2: This is counts per bin, dN, or is it dN/dLogDp? And should you not be plotting dS/dLogDp (surface area rather than number)? The statement that 35% of the overall surface area is being contributed in the gray shaded area is not directly clear when showing only counts. Plotting in surface area space would would best support the stated results about INPs missed by the HINC due to surface area.**

We agree with the reviewer, and changed Figure A2 (revised manuscript) as follows:

Mean particle surface area size distributions at the JFJ during a SDE at 10:00-12:00 UTC on July 10, 2020 (blue line) and between January 1, and December 31, 2020 (black dashed line), both measured with an OAS. Only particles with diameters <  $2.5 \,\mu$ m are sampled by HINC-Auto due to the sampling line geometry and flow rates used. All particle >2.5  $\mu$ m (gray shading) between February 7 and December 31, 2020 contribute during all SDE periods to 35 % to the overall particle surface area. Surface area calculations assumed perfectly spherical particles.

2. Regarding new discussion about including arable soil dusts by altering the term to simply "dust" versus "mineral dust", I am not sure about the term "biological surface features". I understand that this is difficult to generalize, but perhaps biogenic material content? It is not only proteinaceous material being referred to, which I why I might suggest biogenic versus purely biological.

We agree with the reviewer, and changed lines 364-370 (revised manuscript) as proposed:

Whether this is because of the ice nucleation activity of the mineral dust or because of biogenic material content, as proposed by other studies (e.g., see O'Sullivan et al., 2016; Augustin-Bauditz et al., 2016), is

outside of the scope of this study. *Biogenic material* on the dust particles as the predominant cause of the ice nucleation activity of the particles would have two main implications.

Furthermore, we changed lines 372-374 (revised manuscript) as proposed:

Secondly, it would raise the question of the source of the dust particles containing ice-active biogenic material to be potentially from dried lake beds in desert regions or agricultural regions that are not differentiated in this work.

In addition, we changed lines 401-403 (revised manuscript) as proposed:

To validate the stated contribution in future studies or investigate the presence of biogenic material causing the ice-activity, we propose to separate INPs from the bulk aerosol population to analyze the chemical composition of the INPs as well as study the surface using scanning electron microscopy.